# FOAM: Blocked State Folding for Memory-Efficient LLM Training

**Ziqing Wen** [1]  **Jiahuan Wang** [1]  **Ping Luo** [1]  **Dongsheng Li** [1]  **Tao Sun** [1][*]

## Abstract

Large language models (LLMs) have demonstrated remarkable performance due to their large parameter counts and extensive training data. However, their scale leads to significant memory bottlenecks during training, especially when using memory-intensive optimizers like Adam. Existing memory-efficient approaches often rely on techniques such as singular value decomposition (SVD), projections, or weight freezing, which can introduce substantial computational overhead, require additional memory for projections, or degrade model performance. In this paper, we propose Folded Optimizer with Approximate Moment (FOAM), a method that compresses optimizer states by computing block-wise gradient means and incorporates a residual correction to recover lost information. Theoretically, FOAM achieves convergence rates equivalent to vanilla Adam under standard non-convex optimization settings. Empirically, FOAM eliminates up to 90% of the memory overhead of optimizer states and accelerates convergence. Furthermore, FOAM is compatible with other memory-efficient optimizers, delivering performance and throughput that match or surpass both full-rank and existing memory-efficient baselines. Code is available at https://github.com/zqOuO/FOAM.

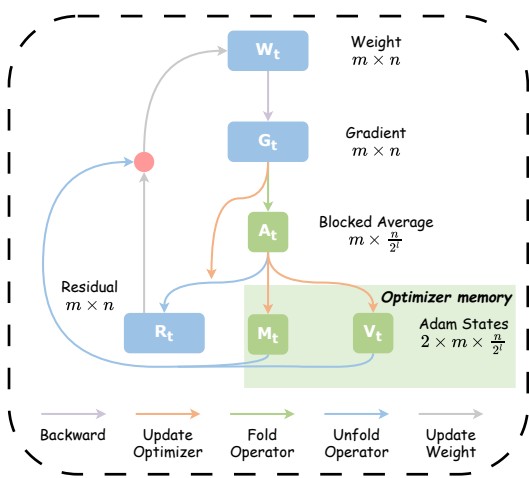

*Figure 1.* Overview of FOAM optimizer with a fold level of $l$.

## 1. Introduction

Large language models (LLMs) have advanced rapidly in recent years and achieved impressive results across a wide range of tasks (Touvron et al., 2023b). This success is primarily driven by vast training datasets and large model sizes (Zhu et al., 2025). As a result, Adam(W) (Kingma & Ba, 2014; Loshchilov & Hutter, 2017) has become the

de facto optimizer for LLM training due to its training efficiency. However, Adam introduces significant memory overhead, consuming twice the model size for storing optimizer states, which makes LLM pre-training and fine-tuning not only compute-intensive but also memory-bound. For instance, even with extremely small training batch sizes, pre-training a 7B model in BF16 still requires at least 58GB of memory (Zhao et al., 2024). For larger models like GPT-3 (Brown et al., 2020), the requirement may exceed 700GB.

To address Adam's memory bottleneck, recent work has focused on algorithmic innovations that optimize the optimizer itself rather than relying solely on hardware or system-level improvements. Unlike system-level approaches—such as checkpointing or quantization (Chen et al., 2016; Dettmers et al., 2021; Zhang et al., 2024b)—which trade off performance or speed for memory savings, algorithmic methods aim to reduce optimizer overhead while preserving full-parameter updates and maintaining Adam's convergence properties. These approaches can be broadly categorized into three classes: (i) reducing the number of trainable parameters by freezing weights, (ii) applying low-rank projections to compress Adam's optimizer states, and (iii) sharing learning rates across blocks to eliminate the need for storing per-parameter second-moment estimates. Representative methods include LoRA (Hu et al., 2022), GaLore (Zhao et al., 2024), and Adam-Mini (Zhang et al., 2024a).

[1]National Key Laboratory of Parallel and Distributed Computing, College of Computer Science and Technology, National University of Defense Technology, Changsha, Hunan, China. Correspondence to: Tao Sun <suntao.saltfish@outlook.com>.

*Proceedings of the 43rd International Conference on Machine Learning*, Seoul, South Korea. PMLR 306, 2026. Copyright 2026 by the author(s).

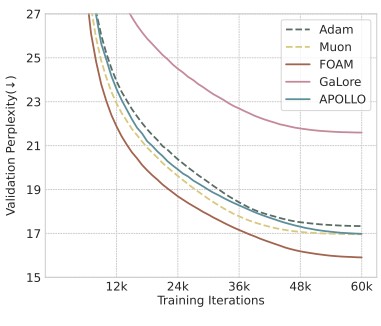 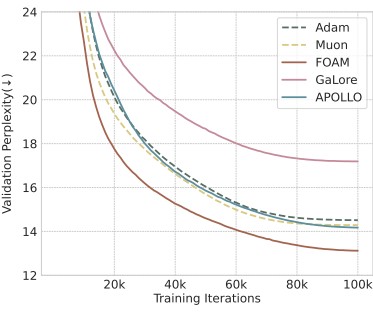 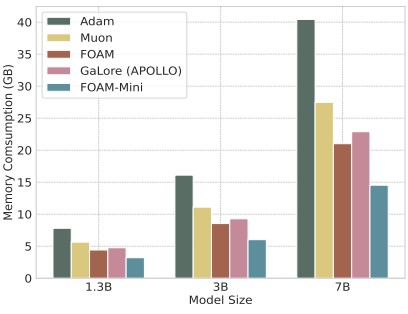

*(a)* 350M model pre-training  *(b)* 1.3B model pre-training  *(c)* End-to-end memory footprint (BF16)

*Figure 2.* **FOAM performance preview on LLM pre-training.** Figure *(a)* and *(b)*: Perplexity learning curves for pre-training LLaMA-350M and 1.3B on C4. FOAM demonstrates superior validation perplexity compared with other baselines. Figure *(c)*: Memory footprint (BF16, model, gradient, and optimizer) for pre-training LLaMA models. FOAM achieves an approximate 50% reduction in memory consumption, and FOAM-Mini further pushes the limit by almost eliminating the memory overhead associated with optimizer states.

LoRA reduces memory consumption by freezing the original pre-trained weights and learning low-rank decompositions of their updates. This design makes it particularly appealing for fine-tuning scenarios with constrained memory resources. However, by restricting parameter updates to a low-rank subspace, LoRA can limit model expressiveness and occasionally degrade performance on certain tasks (Zhang et al., 2023; Xia et al., 2024).

GaLore alleviates this limitation by employing singular value decomposition (SVD) to construct projection matrices that compress Adam's first- and second-moment estimates. While effective, the repeated computation of SVD introduces significant computational overhead, reducing training efficiency. APOLLO (Zhu et al., 2025) mitigates this by replacing SVD with random projections. GWT (Wen et al., 2025) employs wavelet transforms to compress gradients, bypassing the need for SVD. However, under stringent memory constraints, it still causes a substantial reduction in training throughput and a marked performance decline. Adam-Mini (Zhang et al., 2024a) reduces memory usage by sharing second-moment estimates across blocks, eliminating the need for per-parameter second-moment storage. This design caps Adam-mini's achievable optimizer memory efficiency at roughly 50% but limits further reductions in memory overhead for the first-moment estimates.

To overcome these limitations, we propose **FOAM (Folded Optimizer with Approximate Moment)**, a structured state-folding framework that achieves efficient memory compression without sacrificing optimization fidelity. FOAM compresses both moments of Adam by *blocked average*, which preserves structural information at a coarser level, and incorporates a lightweight *residual correction* to recover the information lost during compression, ensuring that updates remain differentiated across parameters. Unlike previous approaches that rely on weight freezing, low-rank

projections, or block-shared learning rates, FOAM supports full-parameter optimization, eliminates the need for projection matrices, simultaneously compresses Adam's first- and second-moment estimates, and is highly effective even under stringent memory constraints. Our main contributions are listed as follows:

- **Innovative Optimizer Design**: We propose FOAM, an efficient memory compression method based on blocked averaging and residual correction. Compared with Full-Adam, FOAM reduces the end-to-end memory footprint up to 50% and the memory overhead of the optimizer states by 90%. Moreover, FOAM can be applied to other memory-intensive optimizers such as Muon (Jordan et al., 2024) and Adam-mini (Zhang et al., 2024a).

- **Theoretical Guarantees**: Within the conventional non-convex optimization framework, we prove that FOAM possesses the same convergence properties as Adam, theoretically ensuring its effectiveness in modern deep learning training tasks.

- **Extensive Evaluation**: We evaluate FOAM on LLM pre-training (LLaMA, Qwen, GPT, DeBERTa models) and fine-tuning (GLUE, MMLU benchmarks) across model sizes (60M–7B) and sequence lengths (256–2048). FOAM outperforms memory-efficient baselines in perplexity and convergence speed without incurring a large computational budget. This makes FOAM a practical, highly efficient, and optimizer-agnostic solution for alleviating the memory bottlenecks in large-scale language model training.

## 2. Related Works

Recent research has explored various algorithmic strategies to reduce the memory overhead of LLM training. Our work

*Table 1.* **Comparison of memory and computational complexity across methods.** Let $W \in \mathbb{R}^{m \times n}$ with $m \leq n$. GaLore and APOLLO use a rank $r$, whereas FOAM uses a level $l$.

|  | Adam | Muon | Adam-Mini | GaLore | APOLLO | FOAM | FOAM-Mini |
|---|---|---|---|---|---|---|---|
| Optimizer Memory | $2mn$ | $mn$ | $mn$ | $mr + 2nr$ | $mr + 2nr$ | $mn/2^{l-1}$ | $2m$ |
| Complexity | $\mathcal{O}(mn)$ | $\mathcal{O}(m^2n)$ | $\mathcal{O}(mn)$ | $\mathcal{O}(m^2n)$ | $\mathcal{O}(mnr)$ | $\mathcal{O}(mn)$ | $\mathcal{O}(mn)$ |

follows this line of inquiry, aiming to enhance memory efficiency through optimizer and state-level innovations.

A prominent direction focuses on reducing the number of trainable parameters. LoRA (Hu et al., 2022) achieves this by freezing the pre-trained model weights and learning low-rank decompositions of their updates, substantially lowering memory usage during fine-tuning. The approach has inspired numerous extensions and refinements, including Batched LoRA (Wen & Chaudhuri, 2023), ReLoRA (Lialin et al., 2023), FLORA (Hao et al., 2024), DoRA (Liu et al., 2024), and BlockedLLM (Ramesh et al., 2024).

Another major line of work targets the substantial memory footprint of optimizer states, particularly in Adam. These methods project gradients and perform updates within a low-dimensional subspace, thereby reducing the need to store high-dimensional moment estimates. Representative examples include Adafactor (Shazeer & Stern, 2018), Ga-Lore (Zhao et al., 2024), Fira (Chen et al., 2024), GWT (Wen et al., 2025), APOLLO (Zhu et al., 2025), LDAdam (Robert et al., 2025), and SubTrack++ (Rajabi et al., 2025). Both Fira, GWT, APOLLO, LDAdam, and SubTrack++ have demonstrated competitive or superior performance compared to full-rank Adam, underscoring the effectiveness of subspace-based optimization.

Other approaches, such as Adam-Mini (Zhang et al., 2024a) and SGDSal (Xu et al., 2024), reduce memory consumption by employing learning rate sharing schemes. AdamS (Zhang et al., 2025) leverages momentum to introduce a normalizer. Meanwhile, Muon (Jordan et al., 2024; Liu et al., 2025) and Scion (Pethick et al., 2025) incorporate orthogonalized gradients, enabling SGD-style updates.

## 3. Motivation and Algorithm

In this section, we first present the detailed procedure for the Adam(W) (Kingma & Ba, 2014; Loshchilov & Hutter, 2017), followed by introducing our **Folded Optimizer with Approximate Moments (FOAM)**.

### 3.1. Full-Adam

At each time step $t$, the Adam optimizer updates the weight matrix $W \in \mathbb{R}^{m \times n}$ by leveraging the first-order moment

and the second-order moment by

$$W_{t+1} = W_t - \eta_t \cdot \frac{M_t}{\sqrt{V_t + \epsilon}}, \qquad (1)$$

where $\eta_t > 0$ denotes the step size (i.e., the learning rate) at time step $t$, and $\cdot$ denotes element-wise multiplication. Here, $M_t, V_t \in \mathbb{R}^{m \times n}$, denote the first and second-order moment, and $\epsilon > 0$ for numerical stability. The moments $M_t$ and $V_t$ are updated by

$$\begin{aligned} M_t &= \beta_1 \cdot M_{t-1} + (1 - \beta_1) \cdot G_t, \\ V_t &= \beta_2 \cdot V_{t-1} + (1 - \beta_2) \cdot G_t^2, \end{aligned} \qquad (2)$$

where $G_t \in \mathbb{R}^{m \times n}$ denotes the stochastic gradient at step $t$, and $\beta_1, \beta_2 \in [0, 1)$ are the decay rates. Adam stores both $M_t$ and $V_t$ to implement its adaptive update scheme. This results in an additional memory overhead of $2 \times m \times n$ elements for the optimizer states.

### 3.2. FOAM

The strong performance of the Adam optimizer stems from the first- and second-moment estimates, but this comes at the cost of twice the memory overhead. To address this challenge, we propose FOAM, which compresses both moments by partitioning adjacent gradient entries into blocks and sharing optimizer states within each block. In addition, FOAM introduces a residual correction mechanism to recover information lost due to compression. Specifically, given a gradient $G_t \in \mathbb{R}^{m \times n}$ at time step $t$, and fold-level $l$, FOAM replaces each group of $2^l$ consecutive elements with their mean value for storage (we assume $n \bmod 2^l = 0$, which can be ensured via padding). Specifically, suppose the fold operator matrix $A^{(l)} \in \mathbb{R}^{n \times \frac{n}{2^l}}$ satisfies:

$$A_{i,j}^{(l)} = \begin{cases} \frac{1}{2^l}, & \text{if } (j-1) \cdot 2^l + 1 \leq i \leq j \cdot 2^l \\ 0, & \text{others} \end{cases} \qquad (3)$$

we can obtain the compressed gradient $\tilde{G}_t = G_t A^{(l)} \in \mathbb{R}^{m \times \frac{n}{2^l}}$, and track the first- and second- moment by

$$\begin{aligned} \tilde{M}_t &= \beta_1 \cdot \tilde{M}_{t-1} + (1 - \beta_1) \cdot \tilde{G}_t \\ \tilde{V}_t &= \beta_2 \cdot \tilde{V}_{t-1} + (1 - \beta_2) \cdot \tilde{G}_t^2 \end{aligned} \qquad (4)$$

To apply the compressed update to the full parameter, we expand it to the original dimension by an *unfold* operator.

Specifically, we define the unfold matrix $E^{(l)} \in \mathbb{R}^{\frac{n}{2^l} \times n}$ as

$$E_{i,j}^{(l)} = \begin{cases} 1, & \text{if } (i-1) \cdot 2^l + 1 \leq j \leq i \cdot 2^l, \\ 0, & \text{otherwise.} \end{cases} \quad (5)$$

This operator replicates each compressed entry across its corresponding block, restoring the update to the full parameter dimension. However, directly replicating the update across dimensions causes all elements within the same block to share identical update values, resulting in weight differences that rely solely on initialization. To overcome this, we introduce the residual $R_t$, which is defined as the fold–unfold operator error at time step $t$

$$R_t = G_t - \tilde{G}_t E^{(l)} = G_t - G_t A^{(l)} E^{(l)}$$

The residual depends solely on the gradient at time step $t$; once computed, it can be released and does not need to be maintained in memory. We introduce the residual term into the expanded estimates of the first-order moment as follows

$$M_t = \tilde{M}_t E^{(l)} + R_t, \quad V_t = \tilde{V}_t E^{(l)} + R_t^2, \quad (6)$$

and update the parameters by Eq. (1). The injection of residual to $M_t$ prevents identical parameter updates within the same block due to broadcasting operations, and ensures that the first-moment estimate $M_t$ includes the complete gradient information $G_t$ of the current step. Similar to Adam-mini, FOAM allows parameters within the same block to share the same second-order moment estimate. The key difference is that FOAM uses a finer-grained block structure and adds an additional residual term to the unfolded second-order moment. We will reveal in Appendix Figure 11 that adding residuals to the second-order moment stabilizes training.

Evidently, FOAM tracks only $1/2^{l-1}$ optimizer state of Full-Adam. We present the pseudocode for FOAM in Algorithm 1. We apply FOAM to the MLP and Attention modules, and Adam to other modules such as Embedding. A scaling coefficient $\alpha$ is introduced to control the ratio of learning rates between FOAM modules and Adam modules. This design is commonly adopted in memory-efficient optimizers (Zhao et al., 2024; Jordan et al., 2024; Zhu et al., 2025).

## 4. Theoretical Results

In this section, we provide the theoretical convergence of FOAM, demonstrating that under the traditional non-convex optimization assumptions, FOAM, even with compressed optimizer states, achieves the same convergence rate as Full-Adam. Before beginning our theoretical analysis, we first collect the following assumptions:

**Assumption 4.1** (*L*-smoothness). Assume the gradient of the loss function $f$ is $L$-smoothness, i.e.,

$$\|\nabla f(W) - \nabla f(W')\| \leq L\|W - W'\|, \forall\, W, W' \in \mathbb{R}^{m \times n}.$$

---

**Algorithm 1** FOAM optimizer

**Input:** 2-D Weight matrix $W$, step size $\{\eta\}_{t \geq 1}$, batch size $b$, decay rates $\beta_1, \beta_2$, iteration $T$, $\epsilon$ for numerical stability, scaling coefficient $\alpha$, fold level $l$, fold and unfold operation $A^{(l)}, E^{(l)}$, and $t = 1$.
**repeat**
    $G_t \leftarrow \frac{1}{b} \sum_{i=1}^{b} \nabla_W f_i(W_t)$       {Batch gradient}
    $\tilde{G}_t \leftarrow G_t A^{(l)}$           {Compress the gradient}
    $R_t \leftarrow G_t - \tilde{G}_t E^{(l)}$       {Compute the residual}
    **if** $t = 1$ **then**
        Initialize $\tilde{M}_0, \tilde{V}_0 \leftarrow 0$
    **end if**

    **Adam states update**
    $\tilde{M}_t \leftarrow \beta_1 \cdot \tilde{M}_{t-1} + (1 - \beta_1) \cdot \tilde{G}_t$
    $\tilde{V}_t \leftarrow \beta_2 \cdot \tilde{V}_{t-1} + (1 - \beta_2) \cdot \tilde{G}_t^2$
    $M_t \leftarrow \tilde{M}_t E^{(l)} + R_t$
    $V_t \leftarrow \tilde{V}_t E^{(l)} + R_t^2$       {Unfold the states}

    $W_t \leftarrow W_{t-1} - \eta_t \cdot \alpha \cdot \frac{M_t}{\sqrt{V_t} + \epsilon}$     {Update weights}
    $t \leftarrow t + 1$
**until** $t = T$
**return** $W_t$

---

**Assumption 4.2** (Bounded variance of the stochastic gradient). Assume that the gradient follows the following noise decomposition

$$G_t = \nabla f(W_t) + \xi_t, \quad \mathbb{E}[\xi_t \mid W_t] = 0, \quad \mathbb{E}\left[\|\xi_t\|^2\right] \leq \sigma^2.$$

**Assumption 4.3** (Bounded gradient). There exists a constant $C \geq 0$ such that

$$\|\nabla f(W_t)\| \leq C, \quad \forall\, t.$$

These assumptions are standard in stochastic optimization and provide the foundation for convergence analysis.

**Theorem 4.4** (Convergence of FOAM). *Let $\{W_t\}_{t \geq 1}$ be generated by Algorithm 1, under the assumptions of Assumptions 4.1 - 4.3, and with $\eta_t = \eta_0/\sqrt{t}$. Then, we have*

$$\min_{1 \leq t \leq T} \mathbb{E}\left[\|\nabla f(W_t)\|^2\right] = \mathcal{O}\left(\frac{\log T + \delta_l^2}{\sqrt{T}}\right) + \mathcal{O}\left(\frac{\sigma^2 \log T}{\sqrt{T}}\right).$$

where $\delta_l = \max_{1 \leq t \leq T} \frac{\|R_t\|}{\|G_t\|}$ denotes the energy ratio of the residual and satisfies $\delta_l \leq 1, \forall\, l$.

This result demonstrates that FOAM achieves the same convergence rate of $\mathcal{O}(\frac{\log T}{\sqrt{T}})$ as Adam (Reddi et al., 2019; Zhou et al., 2024). Furthermore, $\delta_l \leq 1$ for all $l$ does not affect the primary convergence rate, thus theoretically validating the effectiveness of FOAM. In the Appendix Figure 10, we provide the variation of $\frac{\|R_t\|}{\|G_t\|}$ throughout training for different of $l$, where a higher $l$ corresponds to a larger $\delta_l$.

*Table 2.* **Final validation perplexity (lower is better, 3 independent runs for 60M to 350M models) and estimated memory usage (BF16, model, gradient, and optimizer states) for pre-training LLaMA models on the C4 dataset.** FOAM consistently outperforms baselines by achieving comparable or lower validation PPL while requiring significantly less memory.

| Methods | 60M | | 130M | | 350M | | 1.3B | |
|---|---|---|---|---|---|---|---|---|
| | Perplexity | Memory | Perplexity | Memory | Perplexity | Memory | Perplexity | Memory |
| Full-Adam | $29.57_{\pm 0.02}$ | 0.34G | $22.86_{\pm 0.00}$ | 0.80G | $17.33_{\pm 0.00}$ | 2.20G | $14.51_{\pm 0.00}$ | 8.03G |
| Muon | $28.93_{\pm 0.03}$ | 0.30G | $22.75_{\pm 0.01}$ | 0.63G | $16.96_{\pm 0.01}$ | 1.60G | $14.28_{\pm 0.00}$ | 5.61G |
| Adam-Mini | $29.63_{\pm 0.05}$ | 0.22G | $23.73_{\pm 0.01}$ | 0.53G | $17.83_{\pm 0.01}$ | 1.46G | $15.10_{\pm 0.00}$ | 5.35G |
| LDAdamW | $29.27_{\pm 0.07}$ | 0.28G | $22.86_{\pm 0.02}$ | 0.57G | $17.53_{\pm 0.01}$ | 1.38G | $14.88_{\pm 0.00}$ | 4.76G |
| GaLore-1/4 | $34.38_{\pm 0.03}$ | 0.28G | $26.47_{\pm 0.01}$ | 0.57G | $19.36_{\pm 0.02}$ | 1.38G | $15.66_{\pm 0.00}$ | 4.76G |
| APOLLO-1/4 | $31.18_{\pm 0.09}$ | 0.28G | $23.35_{\pm 0.21}$ | 0.57G | $16.73_{\pm 0.05}$ | 1.38G | $14.20_{\pm 0.00}$ | 4.76G |
| FOAM-2 | $\mathbf{28.53}_{\pm 0.01}$ | 0.27G | $\mathbf{22.51}_{\pm 0.00}$ | 0.54G | $\mathbf{15.87}_{\pm 0.00}$ | 1.30G | $\mathbf{13.13}_{\pm 0.00}$ | 4.45G |
| GaLore-1/8 | $39.94_{\pm 1.24}$ | 0.26G | $30.02_{\pm 0.39}$ | 0.52G | $21.59_{\pm 0.23}$ | 1.23G | $17.52_{\pm 0.00}$ | 4.15G |
| APOLLO-1/8 | $31.53_{\pm 0.06}$ | 0.26G | $23.74_{\pm 0.17}$ | 0.52G | $16.98_{\pm 0.10}$ | 1.23G | $14.32_{\pm 0.00}$ | 4.15G |
| FOAM-3 | $\mathbf{28.79}_{\pm 0.05}$ | 0.25G | $\mathbf{22.58}_{\pm 0.03}$ | 0.50G | $\mathbf{15.94}_{\pm 0.02}$ | 1.14G | $\mathbf{13.19}_{\pm 0.00}$ | 3.97G |
| APOLLO-Mini | $31.58_{\pm 0.09}$ | 0.24G | $23.83_{\pm 0.07}$ | 0.46G | $17.17_{\pm 0.06}$ | 1.00G | $14.18_{\pm 0.00}$ | 3.20G |
| GWT-Mini | $32.94_{\pm 0.06}$ | 0.24G | $23.84_{\pm 0.05}$ | 0.46G | $18.12_{\pm 0.03}$ | 1.00G | $14.99_{\pm 0.00}$ | 3.20G |
| FOAM-Mini | $\mathbf{29.71}_{\pm 0.04}$ | 0.24G | $\mathbf{23.10}_{\pm 0.05}$ | 0.46G | $\mathbf{16.53}_{\pm 0.03}$ | 1.00G | $\mathbf{13.43}_{\pm 0.00}$ | 3.20G |
| Training Tokens | 1.3B | | 2.6B | | 7.8B | | 13.1B | |

## 5. Experimental Results

In this section, we experimentally validate the effectiveness of our proposed method, focusing on both pre-training and fine-tuning tasks. For pre-training, we train LLaMA (Touvron et al., 2023a) models of various sizes on the English portion of the C4 (Raffel et al., 2019) dataset. For fine-tuning, we evaluate a range of open-source models on standard downstream tasks. Detailed descriptions of experimental settings and computational environments can be found in the Appendix. Throughout our experiments, we use FOAM-2 to denote the Adam optimizer with $l = 2$. We also define FOAM-Mini as a variant of FOAM that applies $l = \lfloor \log_2 \dim_{\text{hidden}} \rfloor$, resulting in memory consumption approximately equivalent to that of rank-1 methods.

### 5.1. Memory-Efficient Pre-Training

We demonstrate that our proposed method, FOAM, is highly effective for the pre-training of LLMs, achieving superior performance in terms of validation perplexity (PPL), training throughput, memory efficiency, and convergence speed across a range of LLaMA model sizes. Notably, FOAM-c—a variant with an extremely low memory footprint for optimizer states—performs on par with both full-rank baselines and existing memory-efficient methods.

**Training Configuration.** All pre-training experiments are conducted using the BF16 format to reduce memory consumption. We adopt the LLaMA model configurations provided by Zhao et al. (2024). Following prior work, we use a maximum sequence length of 256 and a batch size of 512,

yielding a total of 131K tokens per batch. The learning rate is linearly warmed up during the first 10% of training steps and decayed thereafter using a cosine schedule.

**Baselines.** To ensure a comprehensive comparison, we reproduce results for several representative optimizers. These include: **Full-Adam** (Kingma & Ba, 2014), the standard optimizer for training large models. **Muon** (Liu et al., 2025), an SGD-momentum variant with Newton-Schulz (N-S) iterations; **Adam-Mini** (Zhang et al., 2024a), Adam with block-shared second-moment estimate; **GaLore** (Zhao et al., 2024), a memory-efficient Adam with low-rank gradient projections; **LDAdamW** (Robert et al., 2025), a GaLore variant with continually rotating and persistent error feedback; **APOLLO** (Zhu et al., 2025), which reduces memory usage via random projections. **GWT** (Wen et al., 2025), which reduces memory by wavelet transforms.

For Full-Adam, Muon, and Adam-Mini optimizers, we report the best results from a learning rate sweep over $\{1e-4, 2.5e-4, 5e-4, 1e-3, 2.5e-3, 5e-3, 1e-2\}$. For GaLore, APOLLO, GWT, FOAM and FOAM-Mini, we report the best results from a learning rate sweep over $\{1e-3, 2.5e-3, 5e-3, 1e-2\}$ and choose the $\alpha$ in $\{0.25, 0.5, 1.0\}$. In particular, GaLore-1/8 refers to the configuration where the projection rank is one-eighth of the hidden dimension—corresponding to the memory footprint of FOAM with 3-level gradient folding.

**Main Reults.** We evaluate the performance of FOAM and FOAM-Mini by comparing their memory overhead (calculated layer-by-layer, with details in the Appendix) and final

*Table 3.* **Pre-training LLaMA-3B and LLaMA-7B models on the C4 dataset.** We report the validation PPL across training steps, wall-clock training time, and real memory consumption with gradient checkpointing during training. We train LLaMA-3B on 16 NVIDIA RTX 3090 GPUs (24GB each), and LLaMA-7B on 4 NVIDIA H100 GPUs (80GB each).

| Models | Methods | Memory | 30k | 60k | 90k | 120k | 150k | Time (h) | Tokens/s |
|---|---|---|---|---|---|---|---|---|---|
| | 8-bit Adam | 5.30G | 18.63 | 15.69 | 14.36 | 14.19 | - | 130.1 | 34.5K |
| | Muon | 5.47G | 18.14 | 15.40 | 14.13 | 14.02 | - | 297.3 | 16.3K |
| LLaMA-3B | GaLore-1/4 | 3.91G | 18.44 | 15.98 | 14.90 | 14.73 | - | 143.6 | 35.7K |
| | APOLLO-1/4 | 3.91G | 19.49 | 15.40 | 14.09 | 13.75 | - | 125.6 | 35.9K |
| | FOAM-2 | 3.20G | 15.71 | 13.46 | 12.29 | **11.98** | - | 126.2 | 35.7K |
| | 8-bit Adam | 13.5G | 18.73 | 16.13 | 14.56 | 13.40 | 13.23 | 285.5 | 19.9K |
| | Muon | 14.0G | 18.16 | 15.71 | 13.87 | 13.23 | 13.19 | 350.0 | 16.0K |
| LLaMA-7B | GaLore-1/4 | 9.40G | 18.35 | 15.63 | 14.33 | 13.77 | 13.69 | 341.8 | 20.9K |
| | APOLLO-1/4 | 9.40G | 18.49 | 15.17 | 13.54 | 12.80 | 12.63 | 270.1 | 21.1K |
| | FOAM-2 | 7.52G | 15.78 | 13.33 | 12.01 | 11.39 | **11.13** | 270.2 | 21.0K |

validation PPL against various baseline optimizers. The final PPL results for LLaMA models ranging from 60M to 1.3B parameters are summarized in Table 2. Results for pre-training LLaMA-3B and LLaMA-7B with gradient checkpointing (Chen et al., 2016) are presented in Table 3. Additionally, we include the PPL learning curves for LLaMA-350M and 1.3B in Figures 2a and 2b, for LLaMA-60M and 130M in Figure 6, and the real training memory consumption for LLaMA-1.3B in Appendix Table 14. These results lead to several key observations:

**FOAM Demonstrates Superior Effectiveness in Pre-training:** Under varying memory constraints (i.e., $d_{model}/4$, $d_{model}/8$, and rank 1), FOAM consistently outperforms all other memory-efficient optimizers in final validation PPL, even surpassing Full-Adam and Muon with tuned learning rates, while maintaining significantly lower memory usage. Specifically, for the LLaMA-1.3B model, FOAM-3 consumes approximately 3.97 GB of memory, reducing optimizer memory overhead by 79% and total memory overhead by 50% compared to Adam. It also achieves a 20% reduction in memory usage relative to GaLore and APOLLO, both configured at $1/4$, $d_{model}$ in the 7B model. Additionally, FOAM-Mini decreases optimizer memory overhead by 90%, while maintaining performance comparable to Adam.

**Faster Convergence and Higher Throughput:** As shown in Figures 2a and 2b, FOAM achieves a 2× speedup in convergence steps, outperforming or matching other memory-efficient optimizers in terms of PPL. For larger models, such as LLaMA-3B and LLaMA-7B (as presented in Table 3), FOAM continues to outperform the tested baselines in both convergence speed and final validation PPL, while matching the training throughput of SVD-free APOLLO.

## 5.2. Memory-Efficient Fine-Tuning

In this section, we further evaluate FOAM in LLM fine-tuning, a widely adopted practice in both academia and industry. Our experiments include fine-tuning RoBERTa-Large (Liu, 2019) on the GLUE benchmark (Wang et al., 2018) and adapting several open-source models for evaluation on the MMLU benchmark (Hendrycks et al., 2020).

**Training Configuration.** For GLUE fine-tuning, we fine-tune each task for 3 epochs. For MMLU tasks, we employ 3 open-source models: Gemma3-1B (Team et al., 2025), LLaMA3.2-3B (Grattafiori et al., 2024), and Qwen2.5-7B (Yang et al., 2024) with a sequence length of 2048. Detailed experimental settings are provided in the Appendix.

*Table 4.* **Evaluating FOAM for memory-efficient fine-tuning on the MMLU benchmark** (1 A100 40GB GPU). We report the best average accuracy obtained by sweeping the learning rate over the range {1e−5, 2.5e−5, 5e−5, 1e−4, 1.5e−4, 2e−4}.

| Models | Methods | STEM | Soc. | Hum. | Other | Avg. |
|---|---|---|---|---|---|---|
| | Full-Adam | 27.63 | 26.84 | 25.12 | 25.14 | 26.04 |
| | LoRA | 26.61 | 25.93 | 24.78 | 25.02 | 25.48 |
| Gemma3-1B | GaLore | 27.40 | 26.71 | 24.80 | 25.69 | 25.99 |
| | APOLLO | 27.47 | 27.14 | 25.61 | 25.76 | **26.38** |
| | FOAM | 26.01 | 26.39 | 25.44 | 26.40 | 25.99 |
| | Full-Adam | 46.55 | 65.81 | 49.16 | 63.17 | 55.48 |
| | LoRA | 47.22 | 63.41 | 49.56 | 61.57 | 54.86 |
| LLaMA3.2-3B | GaLore | 46.62 | 65.58 | 48.37 | 63.33 | 55.22 |
| | APOLLO | 46.16 | 65.91 | 49.03 | 62.80 | 55.29 |
| | FOAM | 46.72 | 65.19 | 49.29 | 62.77 | **55.33** |
| | Full-Adam | | OOM | | | - |
| | LoRA | 67.69 | 82.91 | 66.29 | 75.72 | 72.41 |
| Qwen2.5-7B | GaLore | 68.32 | 82.87 | 66.08 | 76.10 | 72.55 |
| | APOLLO | 68.49 | 82.97 | 65.80 | 76.65 | 72.65 |
| | FOAM | 69.12 | 83.49 | 66.48 | 76.28 | **73.04** |

**Baselines.** Consistent with the pre-training experiments, we compare FOAM against Full-Adam, GaLore, and APOLLO. Additionally, we include **LoRA** (Hu et al., 2022) as a baseline. All methods adopt the same hyperparameter strategy to ensure fair comparisons. FOAM 's level $l$ is set to 8 for both GLUE and MMLU tasks, except for Gemma3-1B where $l = 7$. For the other baselines, the rank is set to 4 on GLUE and 8 on MMLU, ensuring comparable mem-

*Table 5.* **Evaluating `FOAM` for memory-efficient fine-tuning on the GLUE benchmark (higher is better)**, using a pre-trained RoBERTa-Large model. We report overall (matched and mismatched) accuracy for MNLI, Matthew's correlation coefficient for CoLA, Pearson correlation for STS-B, and classification accuracy for all other tasks.

| Methods | Memory | CoLA | STS-B | MRPC | RTE | SST2 | MNLI | QNLI | QQP | Avg. |
|---|---|---|---|---|---|---|---|---|---|---|
| Full-Adam | 2.13G | 64.85 | 91.60 | 92.79 | 78.81 | 96.44 | 90.51 | 94.43 | 91.90 | 87.66 |
| LoRA | 0.73G | **64.32** | 90.68 | 91.39 | 77.72 | 95.98 | **90.57** | **94.78** | **90.93** | 87.04 |
| GaLore | 0.72G | 62.52 | 91.18 | 90.94 | 77.11 | **96.10** | 90.27 | 94.21 | 89.99 | 86.54 |
| APOLLO | 0.72G | 61.13 | 91.66 | 92.14 | 80.14 | 95.06 | 89.65 | 93.61 | 89.27 | 86.58 |
| FOAM | 0.71G | 64.30 | **92.44** | **92.33** | **83.03** | 95.75 | 89.74 | 93.88 | 89.65 | **87.64** |

ory usage. Since the level $l$ is already small, we don't run `FOAM-Mini`.

**Main Results.** As reported in Tables 4 and 5, `FOAM` achieves comparable performance relative to baselines across a range of downstream LLM tasks. These results highlight `FOAM`'s effectiveness beyond pre-training and suggest it can serve as a unified, memory-efficient optimization method applicable throughout the LLM training.

# 6. Extra Investigation and Ablation Study

In this section, we present additional empirical studies to further evaluate the robustness and generalization of `FOAM`. Our analysis includes: (i) Pre-training Qwen2.5 (Yang et al., 2024) models at multiple scales; (ii) Assessing `FOAM` under long-sequence training regimes and evaluating its performance with large total training-token budgets; (iii) Ablation study of the hyperparameter $l$; (iv) an ablation study of the residual term in Eq. (6); (v) Evaluation `FOAM` under 8-bit quantization. Beyond these core experiments, the Appendix includes additional studies on GPT (Radford et al., 2019) and DeBERTa (He et al., 2021) models, a further ablation of the scaling coefficient $\alpha$, and a comparison of `FOAM` with Adam using a similar module-wise learning rate design. These results demonstrate that `FOAM` is robust to the hyperparameter $\alpha$ and achieves performance on par with Adam when using the same modul learning rate design.

*Table 6.* Final validation PPL for pre-training Qwen models on C4.

| Methods | 60M | 130M | 350M |
|---|---|---|---|
| Full-Adam | 29.71 | 22.82 | 16.97 |
| Adam-mini | 30.04 | 24.12 | 17.55 |
| Muon | 28.93 | 22.16 | 16.78 |
| GaLore-1/4 | 33.22 | 25.63 | 19.58 |
| APOLLO-1/4 | 30.00 | 23.43 | 16.83 |
| FOAM-2 | **28.57** | **21.25** | **15.80** |
| APOLLO-Mini | 32.05 | 23.34 | 16.97 |
| GWT-Mini | 31.88 | 23.31 | 17.60 |
| FOAM-Mini | **27.98** | **21.75** | **15.99** |

**Pre-training Qwen on C4.** To assess the generalization capability of `FOAM` across different model families, we also pre-train Qwen2.5 (Yang et al., 2024) models on the C4 dataset, following the same experimental setup in LLaMA experiments. Model sizes range from 60M to 350M parameters, and the final validation PPL results are summarized in Table 6. Because these model sizes align with those in Table 2, their estimated memory footprints are similar. As shown in the results, both `FOAM` and `FOAM-Mini` consistently outperform all baseline optimizers, demonstrating that `FOAM` generalizes effectively beyond LLaMA architectures.

*Table 7.* Performance comparison under different sequence lengths.

| Sequence length | Methods | 60M | 130M | 350M |
|---|---|---|---|---|
| 512 | Full-Adam | 31.54 | 23.77 | 17.98 |
| | Muon | 30.36 | 23.48 | 17.62 |
| | Adam-mini | 31.91 | 24.77 | 18.91 |
| | GaLore-1/4 | 35.25 | 27.19 | 19.92 |
| | APOLLO-1/4 | 32.02 | 24.04 | 17.26 |
| | FOAM-2 | **29.26** | **22.47** | **16.34** |
| 1024 | Full-Adam | 35.07 | 26.34 | 20.72 |
| | Muon | 33.21 | 26.11 | 19.60 |
| | Adam-mini | 36.30 | 28.41 | 21.47 |
| | GaLore-1/4 | 38.09 | 29.51 | 21.73 |
| | APOLLO-1/4 | 34.04 | 25.93 | 18.77 |
| | FOAM-2 | **31.69** | **24.38** | **17.75** |

**Long-context Training.** Long-context training is essential for improving the contextual reasoning capabilities of LLMs. To assess `FOAM`'s generalization and efficiency under extended context windows, we pre-train LLaMA models on the C4 dataset using a range of sequence lengths. In all settings, the total number of tokens per batch is kept fixed to ensure a consistent computational budget and comparable token throughput. As shown in Table 7, although all optimizers experience some degradation in PPL as the sequence length increases, `FOAM` consistently achieves the best performance. These results demonstrate `FOAM`'s robustness and stability when training with long input sequences.

**Impact of the `FOAM` Level $l$.** We perform an ablation study to assess how the `FOAM` level $l$ influences performance. For comparison, we include GaLore, APOLLO,

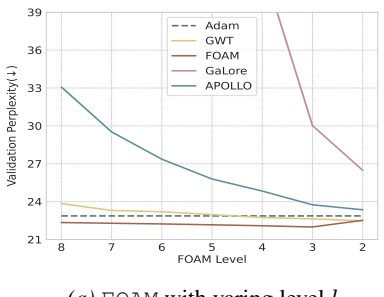

*(a)* FOAM with varing level $l$.

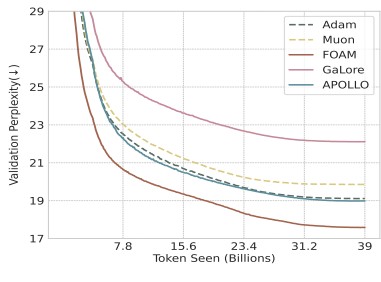

*(b)* Overtraining LLaMA-130M.

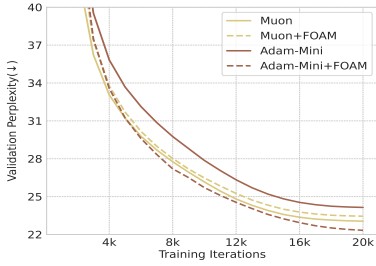

*(c)* FOAM with Adam-Mini and Muon.

*Figure 3.* **Additional Investigation of FOAM.** *(a)* Impact of the FOAM level $l$: FOAM exhibits strong robustness across varying memory constraints. *(b)* Extended training of LLaMA-130M on 39B tokens. *(c)* Integration of FOAM with Adam-Mini and Muon.

and GWT as baselines, configuring their projection ranks and compression levels to roughly match FOAM's memory usage under each setting. As illustrated in Figure 3a, Ga-Lore, APOLLO, and GWT exhibit notable degradation in final validation PPL when operating at lower memory consumptions (corresponding to higher $l$ values). In contrast, FOAM remains remarkably stable: its performance is largely insensitive to variations in $l$, consistently outperforming Adam even under aggressive memory budgets. These results highlight FOAM's robustness and its ability to maintain high optimization quality across a wide range of fold levels.

**Overtraining LLaMA on C4.** Because FOAM, GaLore, and APOLLO generally use larger default learning rates than Full-Adam, they may be more susceptible to instability during prolonged training. To assess FOAM's resilience in such overtraining regimes, we extend the training of LLaMA-130M to 39B tokens-approximately 300 tokens per parameter, roughly 15× the Chinchilla compute allocation (Hoffmann et al., 2022). As shown in Figure 3b, FOAM remains stable and continues to improve, demonstrating strong robustness even under extreme training durations.

**Integration of FOAM with Adam-Mini and Muon.** We further show that FOAM can be seamlessly combined with other optimizers and memory-saving techniques. To illustrate this flexibility, we integrate FOAM into Adam-Mini (Zhang et al., 2024a) and Muon (Liu et al., 2025), and pre-train a LLaMA-130M model. The resulting PPL learning curves, shown in Figure 3c, indicate that FOAM consistently matches or exceeds the performance of the original optimizers. These results highlight FOAM's versatility and effectiveness as a plug-and-play, memory-efficient optimization solution applicable across diverse optimization strategies.

**Impact of the Residual.** We conduct an ablation study on the residual term in Eq. (6) by pre-training the LLaMA-60M and 130M models with and without $R_t$. In addition, we measure the cosine similarity between the update magnitudes produced by FOAM and those of Adam, comparing versions with and without the residual term (Appendix Figure 5). We present the PPL learning curves in Figure 4. When the residual term is omitted, all parameters within the same block receive identical updates, with any intra-block differences attributable solely to initialization. Our results show that including the residual term significantly accelerates convergence and yields lower final validation PPL, underscoring its critical role in ensuring effective optimization with FOAM.

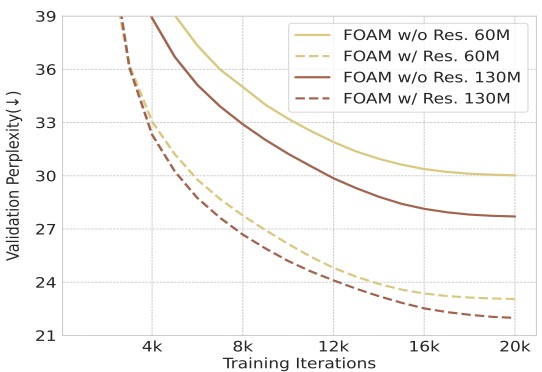

*Figure 4.* Validation PPL with or without residual.

*Table 8.* Evaluating FOAM under 8-bit state quantization.

| Methods | 60M | | 130M | | 350M | |
|---|---|---|---|---|---|---|
| | PPL | Mem. | PPL | Mem. | PPL | Mem. |
| 8-bit-Adam | 30.78 | 0.22G | 23.30 | 0.53G | 17.99 | 1.46G |
| 8-bit-GaLore | 34.88 | 0.18G | 25.53 | 0.38G | 19.79 | 0.92G |
| 8-bit-FOAM-2 | **28.86** | 0.18G | **22.66** | 0.36G | **16.02** | 0.86G |
| Training Tokens | 1.3B | | 2.6B | | 7.8B | |

**FOAM under Int8 Quantization.** Low-bit quantization has become a widely adopted strategy for reducing memory usage in modern LLM training. To assess the robustness of FOAM's compression mechanism under low-precision settings, we incorporate int8-quantized optimizer states

(Dettmers et al., 2022) and pre-train LLaMA models ranging from 60M to 350M. We compare against 8-bit Adam and 8-bit GaLore. Since the low-precision version of APOLLO relies on additional quantization techniques (Zhang et al., 2024b), it is excluded from this evaluation. As shown in Table 8, FOAM maintains superior performance under int8 quantization, demonstrating the resilience and effectiveness of its compression in low-precision regimes.

## 7. Conclusion

In this paper, we propose FOAM, a memory-efficient training strategy suitable for both pre-training and fine-tuning. FOAM preserves the structural information in the gradient matrix through blocked averaging and reconstructs lost information by a residual correction at each step. This enables high-performance LLM training with reduced memory consumption. Our theoretical analysis shows that FOAM retains the convergence rate of adaptive optimizers. Extensive comparisons with existing methods demonstrate that FOAM effectively reduces memory overhead, accelerates convergence, and improves training speed. Additionally, we show that FOAM is compatible with optimizers beyond Adam. Overall, our approach offers an effective solution to the optimizer-state memory bottleneck in LLM training, providing a complementary alternative to low-rank projection techniques for memory-efficient optimization.

## Impact Statement

This paper presents work whose goal is to advance the field of Machine Learning. There are many potential societal consequences of our work, none of which we feel must be specifically highlighted here.

## Acknowledgements

Tao Sun is supported in part by the National Natural Science Foundation of China (Grant Nos. 62522610, 62376278), and NUDT Foundational Research Funding (JS25-02).

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

# Appendix for

## FOAM: Blocked State Folding for Memory-Efficient LLM Training

## A. Lemmas and Proofs

In this section, we present the proofs of the theorems discussed in the main text. Before beginning our proof, we first establish some useful lemmas to facilitate the process.

### A.1. Additional Lemmas

**Lemma A.1.** *Denote $P^{(l)} = A^{(l)}E^{(l)}$, as $A^{(l)}, E^{(l)}$ defined in Eq. (3) and Eq. (5), we have*

$$\left\| P^{(l)} \right\|_2 = \rho(P^{(l)}) = 1, \text{ and } \left\| I_{n \times n} - P^{(l)} \right\|_2 = \rho(I_{n \times n} - P^{(l)}) = 1,$$

*and*

$$\delta_l = \max_{1 \le t \le T} \frac{\left\| G_t \left( I_{n \times n} - P^{(l)} \right) \right\|}{\|G_t\|} = \max_{1 \le t \le T} \frac{\|R_t\|}{\|G_t\|} \le 1$$

*where $I_{n \times n} \in \mathbb{R}^{n \times n}$ denotes the Identity matrix, and $\rho(\cdot)$ represents the spectral radius.*

*Proof.* By the definition of $E^{(l)}$ and $A^{(l)}$, we have

$$E^{(l)} = 2^l \cdot \left( A^{(l)} \right)^T, \quad E^{(l)} A^{(l)} = I_{n' \times n'}.$$

where $n' = \frac{n}{2^l}$. By the definition of $P^{(l)}$, we have

$$P^{(l)} = A^{(l)} E^{(l)} = 2^l \cdot A^{(l)} \left( A^{(l)} \right)^T, \quad \left( P^{(l)} \right)^T = \left( A^{(l)} E^{(l)} \right)^T = 2^l \cdot \left( A^{(l)} \left( A^{(l)} \right)^T \right)^T = 2^l \cdot A^{(l)} \left( A^{(l)} \right)^T,$$

this proves that $P^{(l)}$ is a symmetric matrix. We consider

$$\left( P^{(l)} \right)^2 = \left( A^{(l)} E^{(l)} \right) \left( A^{(l)} E^{(l)} \right) = A^{(l)} \left( E^{(l)} A^{(l)} \right) E^{(l)} = A^{(l)} I_{n' \times n'} E^{(l)} = A^{(l)} E^{(l)} = P^{(l)},$$

this demonstrates that $P^{(l)}$ is an idempotent matrix. Suppose $\lambda$ is an eigenvalue of $P^{(l)}$ with $\mathbf{v}$ as its associated eigenvector, then we have

$$\left( P^{(l)} \right)^2 \mathbf{v} = P^{(l)} \mathbf{v} \longrightarrow \lambda^2 \mathbf{v} = \lambda \mathbf{v} \longrightarrow \lambda(\lambda - 1) = 0.$$

Therefore, the eigenvalues of $P^{(l)}$ can only be 0 or 1 and $\rho(P^{(l)}) = 1$, this gives us $\|P^{(l)}\|_2 = 1$.

Next, we start to analyse $I_{n \times n} - P^{(l)}$. In the previous proof, we have established that $P^{(l)}$ is a symmetric matrix. Hence, we can see that $I_{n \times n} - P^{(l)}$ is also a symmetric matrix. Consider $\left( I_{n \times n} - P^{(l)} \right)^2$, we have

$$\left( I_{n \times n} - P^{(l)} \right)^2 = (I_{n \times n})^2 - 2 I_{n \times n} P^{(l)} + \left( P^{(l)} \right)^2 = I_{n \times n} - P^{(l)}.$$

This gives us $I_{n \times n} - P^{(l)}$ is an idempotent matrix. Therefore, the eigenvalues of $I_{n \times n} - P^{(l)}$ can only be 0 or 1 and $\rho(I_{n \times n} - P^{(l)}) = 1$. Thus, we can see that

$$\left\| G_t \left( I_{n \times n} - P^{(l)} \right) \right\| \le \|G_t\| \left\| I_{n \times n} - P^{(l)} \right\|_2 \le \|G_t\| \to \delta_l \le 1$$

The proof is completed. □

The following lemma bounds the norm of the difference between the `FOAM` update direction $M_t$ and the gradient $G_t$.

**Lemma A.2.** *Let Assumptions 4.1-4.3 hold, and $\eta_t = \eta_0/\sqrt{t}$. The difference $\Delta_t = M_t - \nabla f(W_t)$ can be decomposed as $\Delta_t = \hat{\Delta}_t + \Xi_t$, where $\mathbb{E}[\Xi_t] = 0$ and $\Xi_t$ is the stochastic noise part. For the deterministic part $\hat{\Delta}_t$, there exist constants $b_1, b_2, b_3, b_4, b_5, b_6$ such that*

$$\mathbb{E}\left[\|\hat{\Delta}_t\|^2\right] \le b_1 \frac{1}{t} + b_2 \beta_1^t + b_3 \beta_1^{2t} + b_4 \frac{\sigma^2}{t} + b_5 \sigma^2 \beta_1^t + b_6 \sigma^2 \beta_1^{2t},$$

*Proof.* From Eq. (4), we have

$$\tilde{M}_t = \beta_1 \cdot \tilde{M}_{t-1} + (1-\beta_1) \cdot \tilde{G}_t = (1-\beta_1) \sum_{j=0}^{t-1} \beta_1^j \tilde{G}_{t-j}.$$

Therefore, by the definition of $R_t$ and the fact $G_t = \nabla f(W_t) + \xi_t$,

$$\Delta_t = M_t - \nabla f(W_t) = \tilde{M}_t E^{(l)} + R_t - G_t + \xi_t$$
$$= \tilde{M}_t E^{(l)} + \left(G_t - G_t A^{(l)} E^{(l)}\right) - G_t + \xi_t = \tilde{M}_t E^{(l)} - G_t A^{(l)} E^{(l)} + \xi_t,$$

and

$$\Delta_t = (1-\beta_1) \sum_{j=0}^{t-1} \beta_1^j \tilde{G}_{t-j} E^{(l)} - \tilde{G}_t E^{(l)} + \xi_t$$

$$= (1-\beta_1) \sum_{j=1}^{t-1} \beta_1^j \tilde{G}_{t-j} E^{(l)} + (1-\beta_1) \tilde{G}_t E^{(l)} - \tilde{G}_t E^{(l)} + \xi_t$$

$$= (1-\beta_1) \sum_{j=1}^{t-1} \beta_1^j \tilde{G}_{t-j} E^{(l)} - \beta_1 \tilde{G}_t E^{(l)} + \xi_t$$

$$= (1-\beta_1) \sum_{j=1}^{t-1} \beta_1^j G_{t-j} P^{(l)} - \beta_1 G_t P^{(l)} + \xi_t. \tag{7}$$

By applying the geometric series summation formula, we obtain

$$\beta_1 G_t P^{(l)} = (1-\beta_1) \sum_{j=1}^{t-1} \beta_1^j G_t P^{(l)} + \beta_1^t G_t P^{(l)}.$$

Substituting the above expression into Eq. (7), we have

$$\Delta_t = (1-\beta_1) \sum_{j=1}^{t-1} \beta_1^j \left(G_{t-j} - G_t\right) P^{(l)} - \beta_1^t G_t P^{(l)} + \xi_t.$$

Given $G_t = \nabla f(W_t) + \xi_t$, this further yields

$$\Delta_t = \underbrace{(1-\beta_1) \sum_{j=1}^{t-1} \beta_1^j \left(\nabla f(W_{t-j}) - \nabla f(W_t)\right) P^{(l)} - \beta_1^t G_t P^{(l)}}_{H_t} + \underbrace{(1-\beta_1) \sum_{j=1}^{t-1} \beta_1^j \left(\xi_{t-j} - \xi_t\right) P^{(l)} + \xi_t}_{\Xi_t}. \tag{8}$$

This gives us

$$\Delta_t = \hat{\Delta}_t + \Xi_t,$$

where

$$\hat{\Delta}_t := (1-\beta_1) \sum_{j=1}^{t-1} \beta_1^j \left(\nabla f(W_{t-j}) - \nabla f(W_t)\right) P^{(l)} - \beta_1^t G_t P^{(l)} \tag{9}$$

By the Assumption 4.2, we have

$$\mathbb{E}\left[(1-\beta_1)\sum_{j=1}^{t-1}\beta_1^j\left(\xi_{t-j}-\xi_t\right)P^{(l)}+\xi_t\right]=\mathbb{E}\left[\Xi_t\right]=0.$$

Next, we consider the upperbound of $\|H_t\|$,

$$\|H_t\|=\left\|(1-\beta_1)\sum_{j=1}^{t-1}\beta_1^j\left(\nabla f(W_{t-j})-\nabla f(W_t)\right)P^{(l)}\right\|=(1-\beta_1)\sum_{j=1}^{t-1}\beta_1^j\left\|\nabla f(W_{t-j})-\nabla f(W_t)P^{(l)}\right\|$$

$$\leq(1-\beta_1)\sum_{j=1}^{t-1}\beta_1^j\|\nabla f(W_{t-j})-\nabla f(W_t)\|\|P^{(l)}\|_2\leq(1-\beta_1)\sum_{j=1}^{t-1}\beta_1^j L\|W_{t-j}-W_t\|, \tag{10}$$

where the last inequality we use the $L$-Lipschitz property in Assumption 4.1 and Lemma A.1. By the recursion in Eq. (1), we have

$$\|W_{t-j}-W_t\|=\left\|\sum_{u=t-j}^{t-1}W_{u+1}-W_u\right\|\leq\sum_{u=t-j}^{t-1}\|W_{u+1}-W_u\|=\sum_{u=t-j}^{t-1}\eta_u\left\|\frac{M_u}{\sqrt{V_u}+\epsilon}\right\| \tag{11}$$

By the recursion of $M_t, V_t$ in Eq. (4), and using the fact that $\mathbb{E}[\|G_t\|^2]=\|\nabla f(W_t)\|^2+\mathbb{E}[\|\xi_t\|^2]\leq C^2+\sigma^2$ from Assumption 4.2 and 4.3, we can bound the second moment of $M_u$ as follows

$$\mathbb{E}[\|M_u\|^2]=\mathbb{E}\left[\left\|(1-\beta_1)\sum_{j=0}^{u-1}\beta_1^j G_{u-j}P^{(l)}+G_u\left(I_{n\times n}-P^{(l)}\right)\right\|^2\right]$$

$$\overset{(a)}{\leq}2(1-\beta_1)^2\mathbb{E}\left[\left\|\sum_{j=0}^{u-1}\beta_1^j G_{u-j}P^{(l)}\right\|^2\right]+2\mathbb{E}\left[|G_u(I_{n\times n}-P^{(l)})|^2\right]$$

$$\overset{(b)}{\leq}2(1-\beta_1)\sum_{j=0}^{u-1}\beta_1^j\mathbb{E}[\|G_{u-j}\|^2]\|P^{(l)}\|_2^2+2\delta_l^2\mathbb{E}[\|G_u\|^2]$$

$$\overset{(c)}{\leq}2(C^2+\sigma^2)+2\delta_l^2(C^2+\sigma^2)=2(1+\delta_l^2)(C^2+\sigma^2), \tag{12}$$

where $(a)$ uses $(a+b)^2\leq2a^2+2b^2$, $(b)$ applies Jensen's inequality to the first term and Lemma A.1 to the second, and $(c)$ follows from the bounded second moment of $G_t$. We denote $C_M=\sqrt{2(1+\delta_l^2)(C^2+\sigma^2)}$ as the upper bound for $\sqrt{\mathbb{E}[\|M_u\|^2]}$.

For the second moment term $V_u$, since $V_u$ is a weighted sum of squared stochastic gradients, we have

$$\mathbb{E}[\|V_u\|^2]=\mathbb{E}\left[\left\|(1-\beta_2)\sum_{j=0}^{u-1}\beta_2^j\left(G_{u-j}P^{(l)}\right)^2+\left(G_u\left(I_{n\times n}-P^{(l)}\right)\right)^2\right\|\right]$$

$$\leq(1-\beta_2)\sum_{j=0}^{u-1}\beta_2^j\mathbb{E}[\|G_{u-j}\|^2]\|P^{(l)}\|_2^2+\mathbb{E}[\|G_u\|^2]\|I_{n\times n}-P^{(l)}\|_2^2$$

$$\leq(C^2+\sigma^2)+\delta_l^2(C^2+\sigma^2)=(1+\delta_l^2)(C^2+\sigma^2). \tag{13}$$

It remains straightforward that for the denominator term

$$\frac{1}{\sqrt{V_t}+\epsilon}\leq\frac{1}{\epsilon}. \tag{14}$$

Substituting Eq. (12) and Eq. (14) into the bound for $\|W_{t-j} - W_t\|$, we obtain the expected distance

$$\mathbb{E}\left[|W_{t-j} - W_t|\right] \leq \sum_{u=t-j}^{t-1} \eta_u \mathbb{E}\left[\left\|\frac{M_u}{\sqrt{V_u} + \epsilon}\right\|\right] \leq \frac{C_M}{\epsilon} \sum_{u=t-j}^{t-1} \eta_u.$$

Substituting the above result into Eq. (10), and taking the expectation, we have

$$\mathbb{E}[\|H_t\|] \leq (1 - \beta_1) \frac{(1 + \delta_l) 2\sqrt{C^2 + \sigma^2} L}{\epsilon} \sum_{j=1}^{t-1} \beta_1^j \sum_{u=t-j}^{t-1} \eta_u$$

$$= (1 - \beta_1) \frac{(1 + \delta_l) 2\sqrt{C^2 + \sigma^2} L}{\epsilon} \sum_{u=1}^{t-1} \eta_u \sum_{j=t-u}^{t-1} \beta_1^j$$

$$= (1 - \beta_1) \frac{2(1 + \delta_l)\sqrt{C^2 + \sigma^2} L}{\epsilon} \sum_{u=1}^{t-1} \eta_u \beta_1^{t-u} \frac{1 - \beta_1^u}{1 - \beta_1}$$

$$= \frac{2(1 + \delta_l)\sqrt{C^2 + \sigma^2} L}{\epsilon} \sum_{u=1}^{t-1} \eta_u \beta_1^{t-u}.$$

Considering the truncation $k = \lfloor \frac{t}{2} \rfloor$, we have

$$\mathbb{E}[\|H_t\|] \leq \frac{(1 + \delta_l) 2\sqrt{C^2 + \sigma^2} L}{\epsilon} \sum_{u=1}^{k-1} \eta_u \beta_1^{t-u} + \frac{(1 + \delta_l) 2\sqrt{C^2 + \sigma^2} L}{\epsilon} \sum_{u=k}^{t-1} \eta_u \beta_1^{t-u}. \tag{15}$$

For the first term,

$$\frac{(1 + \delta_l) 2\sqrt{C^2 + \sigma^2} L}{\epsilon} \sum_{u=1}^{k-1} \eta_u \beta_1^{t-u} \leq \frac{(1 + \delta_l) 2\sqrt{C^2 + \sigma^2} L \eta_0 \beta_1^{\frac{t}{2}}}{(1 - \beta_1)\epsilon}.$$

For the second term in the inequality of Eq. (15),

$$\frac{(1 + \delta_l) 2\sqrt{C^2 + \sigma^2} L}{\epsilon} \sum_{u=k}^{t-1} \eta_u \beta_1^{t-u} \leq \frac{(1 + \delta_l) 2\sqrt{C^2 + \sigma^2} L}{\epsilon} \eta_k \sum_{u=k}^{t-1} \beta_1^{t-u}$$

$$\leq \frac{(1 + \delta_l) 2\sqrt{C^2 + \sigma^2} L}{\epsilon} \frac{1}{\sqrt{\frac{t}{2}}} \frac{\beta_1}{1 - \beta_1}$$

$$= \frac{(1 + \delta_l) 2\sqrt{2}\sqrt{C^2 + \sigma^2} L \beta_1}{(1 - \beta_1)\epsilon\sqrt{t}}.$$

We then have

$$\mathbb{E}[\|H_t\|] \leq \frac{(1 + \delta_l) 2\sqrt{C^2 + \sigma^2} L \eta_0}{(1 - \beta_1)\epsilon} \beta_1^{t/2} + \frac{(1 + \delta_l) 2\sqrt{2}\sqrt{C^2 + \sigma^2} L \beta_1}{(1 - \beta_1)\epsilon} \frac{1}{\sqrt{t}}. \tag{16}$$

For the initialization bias term $\beta_1^t G_t P^{(l)}$ in Eq. (8), we have

$$\mathbb{E}\left[\left\|\beta_1^t G_t P^{(l)}\right\|^2\right] \leq \beta_1^{2t} \mathbb{E}[\|G_t\|^2] \|P^{(l)}\|_2^2 \leq \beta_1^{2t}(C^2 + \sigma^2). \tag{17}$$

Substituting Eq. (16) and Eq. (17) into the definition of $\hat{\Delta}_t$, we obtain

$$\mathbb{E}\left[\|\hat{\Delta} t\|^2\right] = \mathbb{E}\left[\left\|H_t - \beta_1^t G_t P^{(l)}\right\|^2\right]$$

$$\leq 2\mathbb{E}\left[\|H_t\|^2\right] + 2\mathbb{E}\left[\left\|\beta_1^t G_t P^{(l)}\right\|^2\right]$$

$$\leq \frac{32(1 + \delta_l)^2(C^2 + \sigma^2)L^2\beta_1^2}{(1 - \beta_1)^2\epsilon^2} \frac{1}{t} + \frac{16(1 + \delta_l)^2(C^2 + \sigma^2)L^2\eta_0^2}{(1 - \beta_1)^2\epsilon^2}\beta_1^t + 2(C^2 + \sigma^2)\beta_1^{2t}.$$

By rearranging the terms to explicitly separate the impact of the stochastic noise variance $\sigma^2$, we have

$$\mathbb{E}\left[\|\hat{\Delta}_t\|^2\right] \leq b_1 \frac{1}{t} + b_2 \beta_1^t + b_3 \beta_1^{2t} + b_4 \frac{\sigma^2}{t} + b_5 \sigma^2 \beta_1^t + 2\sigma^2 \beta_1^{2t},$$

where the constants are defined as follows:

$$b_1 = \frac{32(1+\delta_l)^2 C^2 L^2 \beta_1^2}{(1-\beta_1)^2 \epsilon^2}, \, b_2 = \frac{16(1+\delta_l)^2 C^2 L^2 \eta_0^2}{(1-\beta_1)^2 \epsilon^2}, \, b_3 = 2C^2,$$

$$b_4 = \frac{32(1+\delta_l)^2 L^2 \beta_1^2}{(1-\beta_1)^2 \epsilon^2}, \, b_5 = \frac{16(1+\delta_l)^2 L^2 \eta_0^2}{(1-\beta_1)^2 \epsilon^2}, \, b_6 = 2.$$

The proof is completed. □

**Lemma A.3.** *Let Assumptions 4.1-4.3 hold, and $\eta_t = \eta_0/\sqrt{t}$. Then, there exist constants $d_1, d_2, d_3$ depend on $L, C, \epsilon$, such that*

$$d_1 \eta_t \mathbb{E}\left[\|\nabla f(W_t)\|\right]^2 \leq \mathbb{E}\left[f(W_t)\right] - \mathbb{E}\left[f(W_{t+1})\right] + d_2 \eta_t \mathbb{E}\left[\|\Delta_t\|^2\right] + d_3 \eta_t^2.$$

*Proof.* By the $L-$smoothness, we have

$$f(W_{t+1}) \leq f(W_t) + \langle \nabla f(W_t), W_{t+1} - W_t \rangle + \frac{L}{2}\|W_{t+1} - W_t\|^2. \tag{18}$$

From the recursion in Eq. (1) and the definition of $\Delta_t$, we have

$$W_{t+1} - W_t = -\eta_t \frac{M_t}{\sqrt{V_t} + \epsilon} = -\eta_t \frac{\nabla f(W_t) + (M_t - \nabla f(W_t))}{\sqrt{V_t} + \epsilon} = -\eta_t \frac{\nabla f(W_t) + \Delta_t}{\sqrt{V_t} + \epsilon}.$$

Substituting into Eq. (18) and taking expectation, we obtain

$$\mathbb{E}\left[f(W_{t+1})\right] \leq \mathbb{E}\left[f(W_t)\right] - \eta_t \mathbb{E}\left[\left\langle \nabla f(W_t), \frac{\nabla f(W_t)}{\sqrt{V_t} + \epsilon} \right\rangle\right]$$

$$- \eta_t \mathbb{E}\left[\left\langle \nabla f(W_t), \frac{\Delta_t}{\sqrt{V_t} + \epsilon} \right\rangle\right] + \frac{L\eta_t^2}{2} \mathbb{E}\left[\left\|\frac{M_t}{\sqrt{V_t} + \epsilon}\right\|^2\right]$$

$$= \mathbb{E}\left[f(W_t)\right] - \eta_t \mathbb{E}\left[\left\langle \nabla f(W_t), \frac{\nabla f(W_t)}{\sqrt{V_t} + \epsilon} \right\rangle\right]$$

$$- \eta_t \mathbb{E}\left[\left\langle \nabla f(W_t), \frac{\hat{\Delta}_t}{\sqrt{V_t} + \epsilon} \right\rangle\right] - \mathbb{E}\left[\left\langle \nabla f(W_t), \frac{\hat{\Xi}_t}{\sqrt{V_t} + \epsilon} \right\rangle\right] + \frac{L\eta_t^2}{2} \mathbb{E}\left[\left\|\frac{M_t}{\sqrt{V_t} + \epsilon}\right\|^2\right].$$

By Lemma A.2 we have

$$\mathbb{E}\left[\left\langle \nabla f(W_t), \frac{\hat{\Xi}_t}{\sqrt{V_t} + \epsilon} \right\rangle\right] = 0.$$

Thus, we have

$$\mathbb{E}\left[f(W_{t+1})\right] \leq \mathbb{E}\left[f(W_t)\right] - \eta_t \mathbb{E}\left[\left\langle \nabla f(W_t), \frac{\nabla f(W_t)}{\sqrt{V_t} + \epsilon} \right\rangle\right]$$

$$- \eta_t \mathbb{E}\left[\left\langle \nabla f(W_t), \frac{\hat{\Delta}_t}{\sqrt{V_t} + \epsilon} \right\rangle\right] + \frac{L\eta_t^2}{2} \mathbb{E}\left[\left\|\frac{M_t}{\sqrt{V_t} + \epsilon}\right\|^2\right]$$

By Eq. (13) and the Assumption 4.2, we have

$$\eta_t \mathbb{E}\left[\left\langle \nabla f(W_t), \frac{\nabla f(W_t)}{\sqrt{V_t} + \epsilon} \right\rangle\right] \geq \eta_t \frac{\|\nabla f(W_t)\|^2}{C_M + \epsilon}.$$

Considering the term $\eta_t \mathbb{E}\left[\left\langle \nabla f(W_t), \frac{\hat{\Delta}_t}{\sqrt{V_t}+\epsilon}\right\rangle\right]$. From Young's inequality, we have for any $r > 0$,

$$\left\langle \nabla f(W_t), \frac{\hat{\Delta}_t}{\sqrt{V_t}+\epsilon}\right\rangle \leq \frac{r}{2}\|\nabla f(W_t)\|^2 + \frac{1}{2r}\left\|\frac{\hat{\Delta}_t}{\sqrt{V_t}+\epsilon}\right\|^2.$$

Let $c_1 = \frac{1}{2C_M+\epsilon}, r = \frac{c_1}{2}$, we have

$$\left\langle \nabla f(W_t), \frac{\hat{\Delta}_t}{\sqrt{V_t}+\epsilon}\right\rangle \leq \frac{c_1}{4}\|\nabla f(W_t)\|^2 + \frac{1}{c_1}\left\|\frac{\hat{\Delta}_t}{\sqrt{V_t}+\epsilon}\right\|^2.$$

Thus,

$$\begin{aligned}
\mathbb{E}\left[\left\langle \nabla f(W_t), \frac{M_t}{\sqrt{V_t}+\epsilon}\right\rangle\right] &= \mathbb{E}\left[\left\langle \nabla f(W_t), \frac{\nabla f(W_t)}{\sqrt{V_t}+\epsilon}\right\rangle\right] + \mathbb{E}\left[\left\langle \nabla f(W_t), \frac{\hat{\Delta}_t}{\sqrt{V_t}+\epsilon}\right\rangle\right] \\
&\geq \mathbb{E}\left[\left\langle \nabla f(W_t), \frac{\nabla f(W_t)}{\sqrt{V_t}+\epsilon}\right\rangle\right] - \left|\mathbb{E}\left[\left\langle \nabla f(W_t), \frac{\hat{\Delta}_t}{\sqrt{V_t}+\epsilon}\right\rangle\right]\right| \\
&\geq c_1\|\nabla f(W_t)\|^2 - \frac{c_1}{4}\|\nabla f(W_t)\|^2 - \frac{1}{c_1}\mathbb{E}\left[\left\|\frac{\hat{\Delta}_t}{\sqrt{V_t}+\epsilon}\right\|^2\right] \\
&= \frac{3c_1}{4}\|\nabla f(W_t)\|^2 - \frac{1}{c_1}\mathbb{E}\left[\left\|\frac{\hat{\Delta}_t}{\sqrt{V_t}+\epsilon}\right\|^2\right],
\end{aligned}$$

Therefore, we can see that

$$\begin{aligned}
\mathbb{E}\left[f(W_{t+1})\right] &\leq \mathbb{E}\left[f(W_t)\right] - \frac{3c_1\eta_t}{4}\mathbb{E}\left[\|\nabla f(W_t)\|^2\right] + \frac{\eta_t}{c_1}\mathbb{E}\left[\left\|\frac{\hat{\Delta}_t}{\sqrt{V_t}+\epsilon}\right\|^2\right] + \frac{L\eta_t^2}{2}\mathbb{E}\left[\left\|\frac{M_t}{\sqrt{V_t}+\epsilon}\right\|^2\right] \\
&\leq \mathbb{E}\left[f(W_t)\right] - \frac{3c_1\eta_t}{4}\mathbb{E}\left[\|\nabla f(W_t)\|^2\right] + \frac{\eta_t}{c_1\epsilon^2}\mathbb{E}\left[\left\|\hat{\Delta}_t\right\|^2\right] + \frac{L\eta_t^2}{2\epsilon^2}\mathbb{E}\left[\|M_t\|^2\right] \\
&\leq \mathbb{E}\left[f(W_t)\right] - \frac{3c_1\eta_t}{4}\mathbb{E}\left[\|\nabla f(W_t)\|^2\right] + \frac{\eta_t}{c_1\epsilon^2}\mathbb{E}\left[\left\|\hat{\Delta}_t\right\|^2\right] + \frac{L\eta_t^2}{2}c_2^2,
\end{aligned}$$

where $c_2$ denotes $\frac{2C_M}{\epsilon}$. By reformulating the inequality above, we obtain

$$\frac{3c_1\eta_t}{4}\mathbb{E}\left[\|\nabla f(W_t)\|^2\right] \leq \mathbb{E}\left[f(W_t)\right] - \mathbb{E}\left[f(W_{t+1})\right] + \frac{\eta_t}{c_1\epsilon^2}\mathbb{E}\left[\left\|\hat{\Delta}_t\right\|^2\right] + \frac{L\eta_t^2}{2}c_2^2.$$

This gives us

$$d_1\eta_t\mathbb{E}\left[\|\nabla f(W_t)\|^2\right] \leq \mathbb{E}\left[f(W_t)\right] - \mathbb{E}\left[f(W_{t+1})\right] + d_2\eta_t\mathbb{E}\left[\|\hat{\Delta}_t\|^2\right] + d_3\eta_t^2,$$

where

$$d_1 = \frac{3}{4(2C_M+\epsilon)}, \quad d_2 = \frac{2C_M+\epsilon}{\epsilon^2}, \quad d_3 = \frac{2LC_M^2}{\epsilon^2}.$$

The proof is completed. □

### A.2. Proof of Theorem 4.4

*Proof.* From Lemma A.2, we have

$$\mathbb{E}\left[\|\hat{\Delta}_t\|^2\right] \leq b_1\frac{1}{t} + b_2\beta_1^t + b_3\beta_1^{2t} + b_4\frac{\sigma^2}{t} + b_5\sigma^2\beta_1^t + b_6\sigma^2\beta_1^{2t}.$$

By summing the above formula from $t = 1$ to $T$ with weight $\eta_t$, we obtain

$$\sum_{t=1}^{T} \eta_t \mathbb{E}\left[\|\hat{\Delta}_t\|^2\right] \leq b_1 \sum_{t=1}^{T} \frac{\eta_t}{t} + b_2 \sum_{t=1}^{T} \eta_t \beta_1^t + b_3 \sum_{t=1}^{T} \eta_t \beta_1^{2t}$$
$$+ b_4 \sigma^2 \sum_{t=1}^{T} \frac{\eta_t}{t} + b_5 \sigma^2 \sum_{t=1}^{T} \eta_t \beta_1^t + b_6 \sigma^2 \sum_{t=1}^{T} \eta_t \beta_1^{2t}.$$

With the fact that $\eta_t = \frac{\eta_0}{\sqrt{t}}$, it is straightforward that

$$\sum_{t=1}^{T} \frac{\eta_t}{t} = \eta_0 \sum_{t=1}^{T} t^{-3/2} = c_3 < +\infty,$$

$$\sum_{t=1}^{T} \eta_t \beta_1^t = \eta_0 \sum_{t=1}^{T} \frac{\beta_1^t}{\sqrt{t}} = c_4 < +\infty,$$

$$\sum_{t=1}^{T} \eta_t \beta_1^{2t} = \eta_0 \sum_{t=1}^{T} \frac{\beta_1^{2t}}{\sqrt{t}} = c_5 < +\infty,$$

$$\sum_{t=1}^{T} \eta_t^2 = \eta_0^2 \sum_{t=1}^{T} \frac{1}{t} \leq \eta_0^2 (1 + \log T).$$

Therefore, we have

$$\sum_{t=1}^{T} \eta_t \mathbb{E}\left[\|\hat{\Delta}t\|^2\right] \leq (b_1 + b_4 \sigma^2) c_3 + (b_2 + b_5 \sigma^2) c_4 + (b_3 + b_6 \sigma^2) c_5.$$

From Lemma A.3, we have

$$d_1 \sum_{t=1}^{T} \eta_t \mathbb{E}\left[\|\nabla f(W_t)\|^2\right] \leq \mathbb{E}\left[f(W_1)\right] - \mathbb{E}\left[f(W_{T+1})\right] + d_2 \sum_{t=1}^{T} \eta_t \mathbb{E}\left[\|\hat{\Delta}t\|^2\right] + d_3 \sum_{t=1}^{T} \eta_t^2$$
$$\leq \mathbb{E}\left[f(W_1)\right] - f^* + d_2 \left[(b_1 + b_4 \sigma^2) c_3 + (b_2 + b_5 \sigma^2) c_4 + (b_3 + b_6 \sigma^2) c_5\right]$$
$$+ d_3 \eta_0^2 (1 + \log T).$$

Dividing both sides of the inequality by $d_1 \sum_{t=1}^{T} \eta_t$, we have

$$\frac{d_1 \sum_{t=1}^{T} \eta_t \mathbb{E}\left[\|\nabla f(W_t)\|^2\right]}{d_1 \sum_{t=1}^{T} \eta_t}$$
$$\leq \frac{\mathbb{E}\left[f(W_1)\right] - f^* + d_2 \left[(b_1 + b_4 \sigma^2) c_3 + (b_2 + b_5 \sigma^2) c_4 + (b_3 + b_6 \sigma^2) c_5\right] + d_3 \eta_0^2 (1 + \log T)}{d_1 \sum_{t=1}^{T} \eta_t}.$$

Using the fact that $\sum_{t=1}^{T} \eta_t \geq 2\eta_0(\sqrt{T+1} - 1)$, and taking the minimum over $t$, we obtain:

$$\min_{1 \leq t \leq T} \mathbb{E}\left[\|\nabla f(W_t)\|^2\right] \leq \frac{\mathbb{E}\left[f(W_1)\right] - f + d_2(b_1 c_3 + b_2 c_4 + b_3 c_5) + d_3 \eta_0^2 (1 + \log T)}{2 d_1 \eta_0 (\sqrt{T+1} - 1)}$$
$$+ \frac{d_2 \sigma^2 (b_4 c_3 + b_5 c_4 + b_6 c_5)}{2 d_1 \eta_0 (\sqrt{T+1} - 1)}.$$

By substituting the definitions of $d_3$ and $b_i$, and noting that $d_3$ and $b_{1,\ldots,6}$ contain the scaling factor $(1 + \delta_l^2)$, we observe that all terms in the numerator are either constants or logarithmic in $T$. Thus, we obtain

$$\min_{1 \leq t \leq T} \mathbb{E}\left[\|\nabla f(W_t)\|^2\right] = \mathcal{O}\left(\frac{\log T + \delta_l^2}{\sqrt{T}}\right) + \mathcal{O}\left(\frac{\sigma^2 \log T}{\sqrt{T}}\right).$$

The proof is completed. $\qquad\square$

## B. Experimental Details

### B.1. Hyperparameters

In this section, we detail the hyperparameters used to reproduce our experimental results. We adopt the hyperparameters of $(\beta_1 = 0.9, \beta_2 = 0.95, \epsilon = 1e{-}8)$ across all the tasks for Adam, as these hyperparameter setting is widely used in LLM training (Touvron et al., 2023a). For pre-training LLaMA models, we fine-tune both Adam (Kingma & Ba, 2014) and Muon (Liu et al., 2025), selecting the learning rate ($lr$) that achieves the lowest PPL from the set $\{1.0e{-}4, 5.0e{-}4, 1.0e{-}3, 2.5e{-}3, 5.0e{-}3, 1.0e{-}2\}$. For GaLore (Zhao et al., 2024) and APOLLO (Zhu et al., 2025), we tune the learning rate within $\{1.0e{-}3, 2.5e{-}3, 5.0e{-}3, 1.0e{-}2\}$. Given that our experimental configuration is similar to that of prior studies, we fine-tune their scaling factors, within $\{0.25, 0.5, 0.75, 1.0\}$ for GaLore, APOLLO, FOAM, FOAM-Mini, and adopt the recommended $\alpha = 128$ for APOLLO-Mini. For our method, both FOAM and FOAM-Mini use a scale factor of $\alpha = 0.25$. All experiments use BF16 precision to reduce memory consumption and are parallelized using Distributed Data Parallel (DDP) across multiple GPUs with gradient synchronization using PyTorch's (Paszke et al., 2017) *torch.distributed* framework.

Following the experimental setups of previous works (Zhao et al., 2024; Zhu et al., 2025), we use a batch size of 512 and a sequence length of 256 by default. FOAM is applied to both the MLP and attention modules (Vaswani et al., 2017). The learning rate is linearly warmed up over the first 10% of iterations, followed by a cosine decay scheduler for the remainder of training.

*Table 9.* Hyperparameters ($lr, \alpha$) for pre-training LLaMA models.

| Models | 60M | | 130M | | 350M | | 1B | | 3B | | 7B | |
|---|---|---|---|---|---|---|---|---|---|---|---|---|
| Hyperparameters | $lr$ | $\alpha$ | $lr$ | $\alpha$ | $lr$ | $\alpha$ | $lr$ | $\alpha$ | $lr$ | $\alpha$ | $lr$ | $\alpha$ |
| Full-Adam (8-bit) | 5.0e-3 | - | 1.0e-3 | - | 1.0e-3 | - | 5.0e-4 | - | 5.0e-4 | - | 5.0e-4 | - |
| Muon | 5.0e-3 | - | 2.5e-3 | - | 1.0e-3 | - | 1.0e-3 | - | 1.0e-3 | - | 5.0e-4 | |
| Adam-mini | 5.0e-3 | - | 1.0e-3 | - | 5.0e-4 | - | 2.5e-4 | - | - | - | - | - |
| LDAdamW | 5.0e-3 | - | 1.0e-3 | - | 1.0e-3 | - | 5.0e-4 | - | | | | |
| GaLore | 1.0e-2 | 0.25 | 1.0e-2 | 0.25 | 1.0e-2 | 0.25 | 1.0e-2 | 0.25 | 5.0e-3 | 0.25 | 1.0e-2 | 0.25 |
| APOLLO | 1.0e-2 | 1.0 | 1.0e-2 | 1.0 | 1.0e-2 | 1.0 | 1.0e-2 | 1.0 | 5.0e-3 | 1.0 | 1.0e-2 | 1.0 |
| GWT | 1.0e-2 | 0.25 | 1.0e-2 | 0.25 | 1.0e-2 | 0.25 | 1.0e-2 | 0.25 | 5.0e-3 | 0.25 | - | - |
| FOAM | 1.0e-2 | 0.25 | 1.0e-2 | 0.25 | 1.0e-2 | 0.25 | 1.0e-2 | 0.25 | 5.0e-3 | 0.25 | 5.0e-3 | 0.25 |
| APOLLO-Mini | 1.0e-2 | 128 | 1.0e-2 | 128 | 1.0e-2 | 128 | 1.0e-2 | 128 | - | - | - | - |
| FOAM-Mini | 1.0e-2 | 0.25 | 1.0e-2 | 0.25 | 1.0e-2 | 0.25 | 1.0e-2 | 0.25 | - | - | - | - |

Additionally, we present a concise summary of the architectural hyperparameters for the LLaMA (Large Language Model Meta AI) (Touvron et al., 2023a), Qwen (Yang et al., 2024), and RoBERTa-Large (Robustly Optimized BERT Approach) (Liu, 2019) models used in the main text. These details are provided in Table 10.

For fine-tuning the RoBERTa-large model (Liu, 2019) on the GLUE benchmark (Wang et al., 2018), we conduct a hyperparameter sweep over the learning rate for each method in the range $\{1e{-}5, 2.5e{-}5, 5e{-}5, 7.5e{-}5, 1e{-}4, 1.5e{-}4, 2e{-}4, 4e{-}4\}$, and report the best performance for each task.

For the MMLU fine-tuning task, we follow the learning rate search strategy proposed in Zhu et al. (2025). Specifically, we evaluate each method by sweeping the learning rate over the range $\{1e{-}5, 2.5e{-}5, 5e{-}5, 1e{-}4, 1.5e{-}4, 2e{-}4\}$, and report the highest average score achieved. A detailed summary of the hyperparameter settings used for fine-tuning FOAM on GLUE is provided in Table 11.

### B.2. Memory Estimation

For memory estimation, we follow the general approach proposed in GaLore (Zhao et al., 2024). Specifically, we isolate memory overhead attributable to model parameters and optimizer states, while excluding other factors such as batch size, PyTorch's memory caching and fragmentation behavior (Paszke et al., 2017), and runtime training configurations.

As a representative example, the LLaMA-60M model contains approximately 58 million parameters. Using BF16 precision

*Table 10.* Architecture hyperparameters of LLaMA for pre-training. Batch size and training data amount are specified in tokens.

| Model | Params | Hidden | Intermediate | Heads | Layers | Iteration | Training tokens |
|---|---|---|---|---|---|---|---|
| LLaMA | 60M | 512 | 1376 | 8 | 8 | 10K | 1.3B |
| | 130M | 768 | 2048 | 12 | 12 | 20K | 2.6B |
| | 350M | 1024 | 2736 | 16 | 24 | 60K | 7.8B |
| | 1B | 2048 | 5461 | 24 | 32 | 100K | 13.1B |
| | 3B | 2560 | 6848 | 32 | 32 | 120K | 15.7B |
| | 7B | 4096 | 11008 | 32 | 32 | 150K | 19.7B |
| Qwen | 60M | 576 | 1536 | 8 | 12 | 10K | 1.3B |
| | 130M | 768 | 2816 | 12 | 14 | 20K | 2.6B |
| | 350M | 1024 | 3328 | 16 | 26 | 60K | 7.8B |
| RoBERTa | 355M | 1024 | 4096 | 16 | 24 | - | - |

*Table 11.* Hyperparameters of fine-tuning RoBERTa-base model on GLUE for FOAM.

| Hyperparameters | CoLA | STS-B | MRPC | RTE | SST2 | MNLI | QNLI | QQP |
|---|---|---|---|---|---|---|---|---|
| Batch Size | 32 | 16 | 16 | 16 | 16 | 16 | 16 | 16 |
| Epochs | | | | 3 | | | | |
| $lr$ Scheduler | | | | Cosine | | | | |
| Warmup Steps | | | | 10% | | | | |
| Where | | | | All | | | | |
| Level $l$ | | | | 8 | | | | |
| Scale $\alpha$ | | | | 0.25 | | | | |
| Max Seq. Len. | | | | 256 | | | | |
| $lr$ | 2.0e-4 | 1.5e-4 | 1.5e-4 | 1.0e-4 | 5.0e-5 | 2.5e-5 | 2.5e-5 | 2.5e-5 |

*Table 12.* Hyperparameters of fine-tuning different models on the MMLU benchmark for FOAM.

| Hyperparameters | Gemma3-1B | LLaMA3.2-3B | Qwen2.5-7B |
|---|---|---|---|
| Scale $\alpha$ | | 0.25 | |
| $lr$ Scheduler | | Cosine | |
| Warmup Steps | | 10% | |
| Epochs | | 3 | |
| Batch Size | | 16 | |
| Where | | All | |
| N-shots | | 5 | |
| Cut-off Len. | | 2048 | |
| Level $l$ | 7 | 8 | 8 |
| $lr$ | 2.0e-4 | 2.5e-5 | 1.0e-5 |

(2 bytes per parameter), this yields a model memory footprint of approximately 0.11 GB. The Adam optimizer maintains two auxiliary states—the first-order moment $M$ and second-order moment $V$—each the same size as the model parameters, resulting in an additional 0.23 GB. The total memory usage with Adam is thus approximately 0.34 GB. For our FOAM method, the computation formula depends on the number of parameters that use FOAM and those that do not. Specifically, for LLaMA-60M, there are 25.3M parameters that use FOAM to save memory, and the remaining 32.77M parameters are

updated using Adam. In the case of using `FOAM` with level $l = 2$, the optimizer states that the parameters are calculated as follows:

- The optimizer state for the `FOAM` parameters:

$$25.3 \text{ M} \times 2 \text{ Bytes} \times 2/4 = 25.3 \text{ MBytes}.$$

- The optimizer state for the Adam parameters:

$$32.77 \text{ M} \times 2 \text{ Bytes} \times 2 = 131.08 \text{ MBytes}.$$

- The memory occupied by the model parameters:

$$116.14 \text{ MBytes}.$$

Therefore, the estimated training memory overhead with `FOAM`-2 is

$$25.3 \text{ MBytes} + 131.08 \text{ MBytes} + 116.14 \text{ MBytes} = 272.52 \text{ MBytes} \approx 0.27 \text{ GBytes}$$

For GaLore and APOLLO, we assume the model weight matrix has shape $m \times n$, where $m < n$. In this case, the optimizer state that includes the projection matrices uses

$$2 \times n \times r + m \times r.$$

If $m > n$, the optimizer state size becomes:

$$2 \times m \times r + n \times r.$$

The total estimated memory usage can be obtained by evaluating the model training code during runtime. Due to the additional projection matrices maintained by the optimizer, the actual memory overhead of GaLore and APOLLO remains higher than that of `FOAM`-2, even when using a rank as low as $1/4$ of the model size. We present the estimated memory consumption in Table 13.

*Table 13.* Estimated model/optimizer states memory comsumption for pre-training 60M-1.3B models.

| Methods | 60M | 130M | 350M | 1B |
|---------|-----|------|------|-----|
| Full-Adam | 0.11G/0.23G | 0.25G/0.51G | 0.68G/1.37G | 2.60G/5.20G |
| Muon | 0.11G/0.19G | 0.25G/0.38G | 0.68G/0.92G | 2.60G/3.61G |
| GaLore-1/4 | 0.11G/0.17G | 0.25G/0.32G | 0.68G/0.70G | 2.60G/2.16G |
| APOLLO-1/4 | 0.11G/0.17G | 0.25G/0.32G | 0.68G/0.70G | 2.60G/2.16G |
| FOAM-2 | 0.11G/0.16G | 0.25G/0.29G | 0.68G/0.62G | 2.60G/1.85G |
| GaLore-1/8 | 0.11G/0.15G | 0.25G/0.27G | 0.68G/0.55G | 2.60G/1.55G |
| APOLLO-1/8 | 0.11G/0.15G | 0.25G/0.27G | 0.68G/0.55G | 2.60G/1.55G |
| FOAM-3 | 0.11G/0.14G | 0.25G/0.25G | 0.68G/0.46G | 2.60G/1.37G |
| APOLLO-Mini | 0.11G/0.13G | 0.25G/0.21G | 0.68G/0.32G | 2.60G/0.60G |
| GWT-Mini | 0.11G/0.13G | 0.25G/0.21G | 0.68G/0.32G | 2.60G/0.60G |
| FOAM-Mini | 0.11G/0.13G | 0.25G/0.21G | 0.68G/0.32G | 2.60G/0.60G |

### B.3. Experiment Enviroments

All pre-training experiments were conducted using 4 to 32 NVIDIA RTX 3090 GPUs and 4 NVIDIA H100 GPUs with PyTorch version 2.3.0. Fine-tuning experiments were performed on a single NVIDIA A100 40GB GPU within the LLaMA-Factory framework (Zheng et al., 2024), using PyTorch version 2.6.0. All experiments use a random seed of 42 for data shuffling.

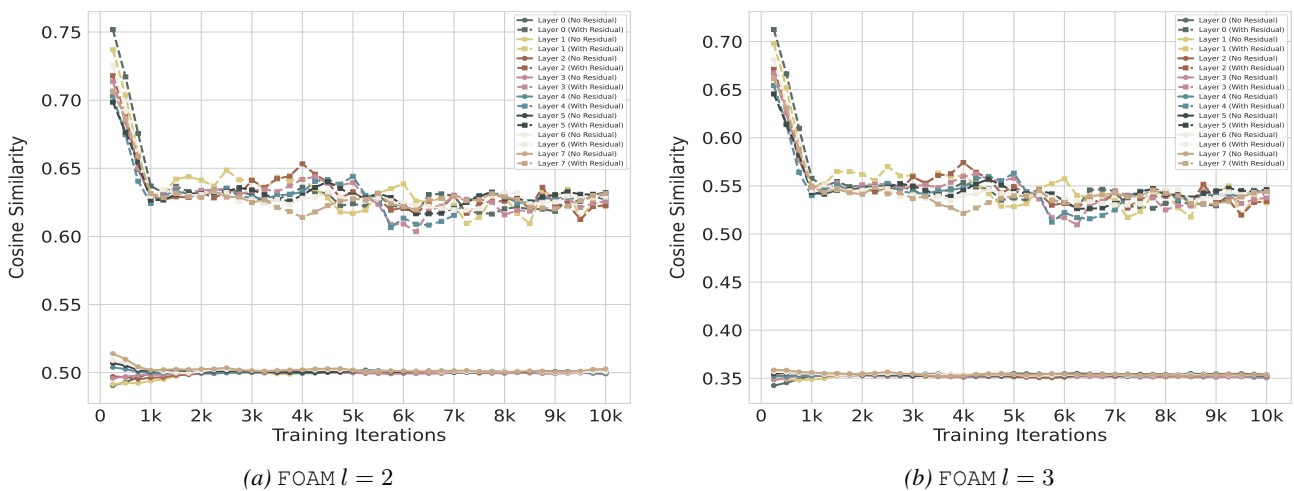

*(a)* FOAM $l = 2$           *(b)* FOAM $l = 3$

*Figure 5.* **Cosine Similarities between the Update Matrices of FOAM with or without Residual and Adam.** We report the average similarity across all modules within each layer. As observed, the update matrices including the residual term exhibit a higher cosine similarity with Adam's updates compared to those without the residual. Specifically, for the setting $l = 3$, FOAM updates maintain a cosine similarity greater than 0.5 with standard Adam, despite retaining only $1/8$ of the original Adam optimizer state.

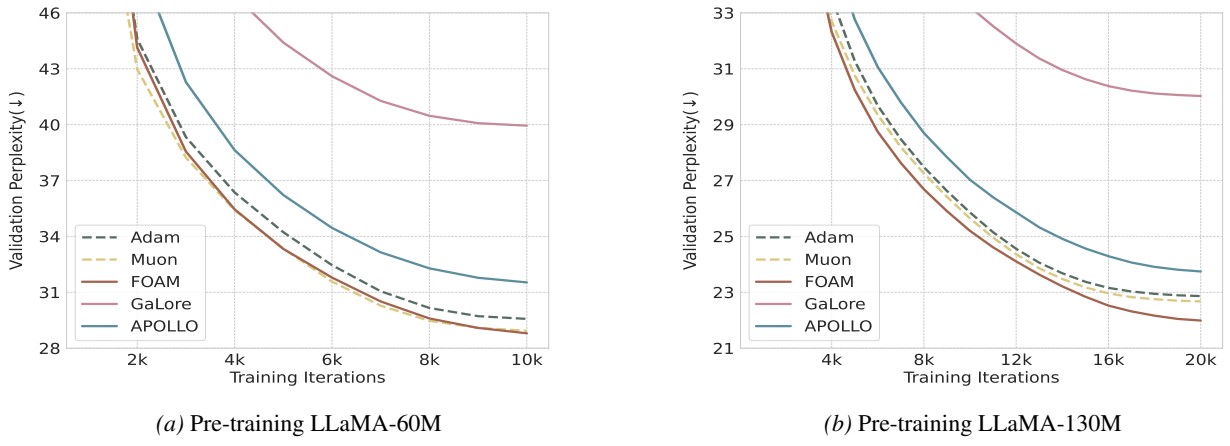

*(a)* Pre-training LLaMA-60M           *(b)* Pre-training LLaMA-130M

*Figure 6.* PPL learning curves of pre-training LLaMA-60M and 130M on C4

## C. Addtional Experiment Results

### C.1. GPT-2 and DeBERTa Experiments

In this section, we test the performance of the FOAM we proposed on GPT-2 (Radford et al., 2019), and DeBERTa (He et al., 2021) models. For each model, we adjust the learning rate within the range {5e-4, 1e-3, 2.5e-3, 5e-3, 1e-2}, keeping the memory-efficient scaling factors unchanged, and train for a total of 20k iterations, covering 2.6B tokens. The experimental results are shown in Figure 7. FOAM continues to achieve leading performance on these models, which demonstrates the scalability of FOAM across other model architectures.

### C.2. Adam with Module-wise Learning Rate

Notably, in this paper, FOAM surpasses Adam on validation metrics across many tasks. Similarly, Fira (Chen et al., 2024) and APOLLO (Zhu et al., 2025) also outperform Adam, even though these memory-efficient methods follow the same hyperparameter tuning strategy as Adam. In this section, we attempt to provide a possible explanation for this phenomenon, namely that it stems from the module-wise optimizer configuration used in current memory-efficient optimizers.

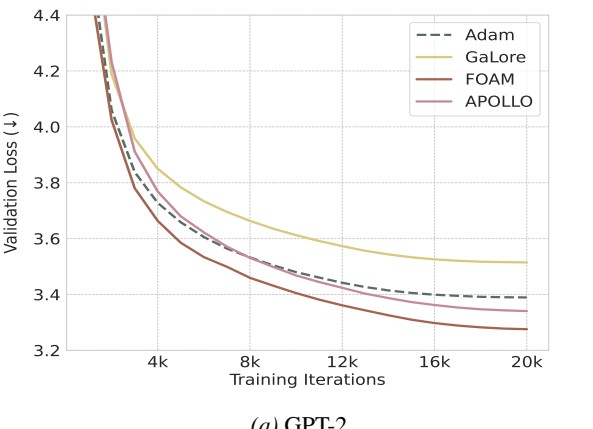
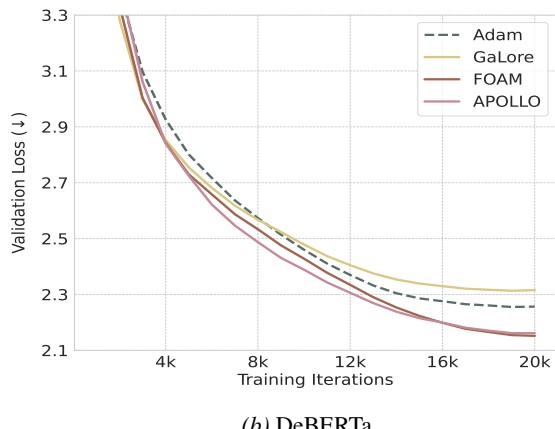

*(a)* GPT-2

*(b)* DeBERTa

*Figure 7.* **Pre-training GPT-2 and DeBERTa models on C4.** *(a)*: Pre-training GPT-2-base model. *(b)*: Pre-training DeBERTa-base model. FOAM continues to achieve leading performance on these models.

Concretely, most current memory-efficient optimizers (Zhao et al., 2024; Jordan et al., 2024; Chen et al., 2024; Zhu et al., 2025) use a hybrid optimizer setup—employing vanilla Adam for modules like Embeddings and LayerNorm, while applying compressed-state optimization to Attention and MLP modules, with a scaling factor $\alpha$ used to adjust the learning rates across modules. For modules such as Embeddings and LayerNorm, the learning rate $lr$ is applied, while Attention and MLP modules use $lr \times \alpha$, effectively creating a module-dependent learning-rate scheme. Prior work (Wang et al., 2025) has shown that Adam with module-wise learning rates converges faster than standard Adam.

Accordingly, to thoroughly assess the FOAM optimizer, we begin with an ablation study on the scaling factor $\alpha$; the experimental results are presented in Figure 8. The results indicate that FOAM is in fact not sensitive to this hyperparameter. In all of our pre-training and fine-tuning experiments, FOAM consistently uses a scaling factor $\alpha = 0.25$, whereas GaLore requires task-specific tuning. This further demonstrates FOAM's robustness to this parameter and the general applicability of this setting.

Furthermore, we compare FOAM with an Adam variant that employs the same block-wise learning-rate scheme, with results presented in Figure 9. The block-wise learning-rate configuration substantially improves Adam's performance. Under this setting, FOAM achieves results comparable to Adam, further demonstrating its effectiveness: with identical hyperparameters, FOAM performs on par with a well-tuned Adam while requiring less memory and achieving higher training throughput.

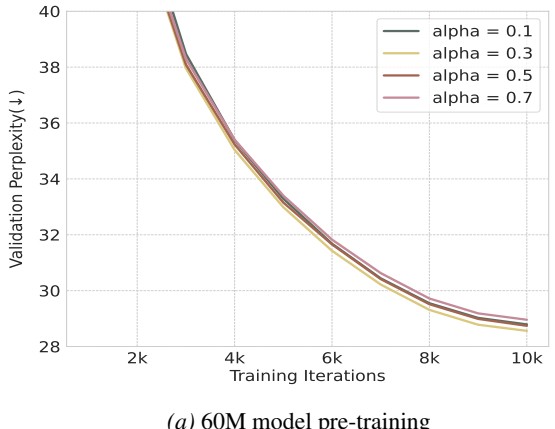
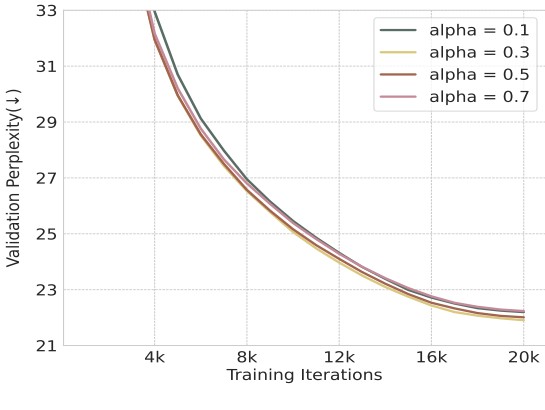

*(a)* 60M model pre-training

*(b)* 130M model pre-training

*Figure 8.* **Study the effects of $\alpha$ in Algorithm 1.** As observed, FOAM is not sensitive to the choice of $\alpha$ in our tests.

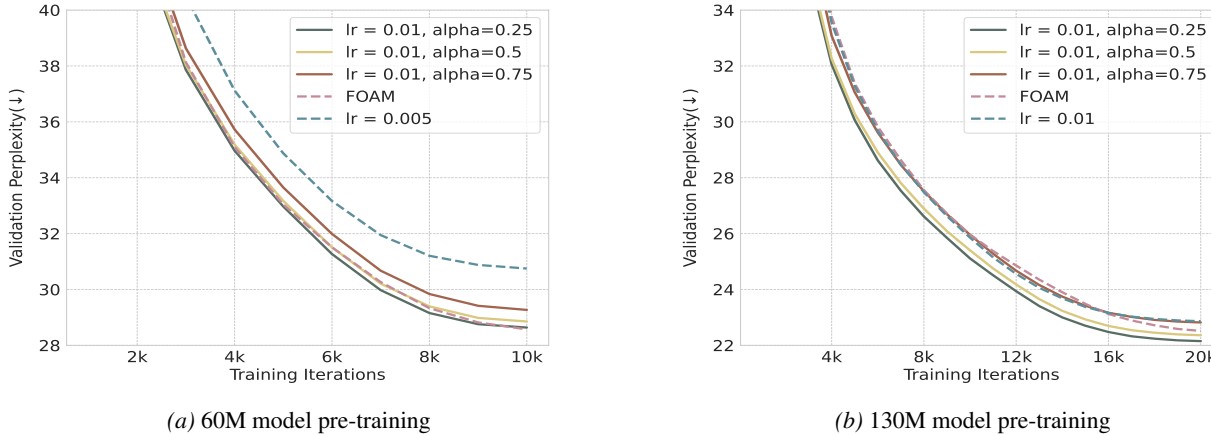

*(a)* 60M model pre-training        *(b)* 130M model pre-training

*Figure 9.* **Comparing FOAM with Adam employing block-wise learning rates.** Block-wise Adam converges more rapidly and yields a lower final validation loss than the uniform-rate variant. FOAM 's performance remains on par with that of block-wise Adam.

## D. Future Works

Several directions remain open for further exploration:

- Due to limitations in computational resources, the maximum model size we used to validate the effectiveness of FOAM was 7B, while experiments on larger models often necessitate more than 100 high-memory GPUs. Thus, the full potential of FOAM for training ultra-large-scale models on significantly larger GPU clusters remains to be elucidated.

- The potential of extending FOAM to encompass additional model architectures, including diffusion models (Song et al., 2021) and Vision Transformers (ViT) (Dosovitskiy et al., 2021). The optimization of such models frequently necessitates the handling of numerous gradients exceeding two dimensions. The adaptation of FOAM to effectively handle these high-dimensional gradients necessitates further architectural design and is thus left for future work.

- Investigating how the FOAM compression strategy influences the overhead of inter-GPU communication. Notably, FOAM only performs compression on neighboring elements, thereby eliminating the need for pre-storing the complete gradient. Consequently, FOAM demonstrates natural compatibility with gradient segmentation methods like ZeRO (Rajbhandari et al., 2020). Since the minimization of inter-GPU communication overhead falls outside the core scope of this manuscript, we reserve this analysis for subsequent research.

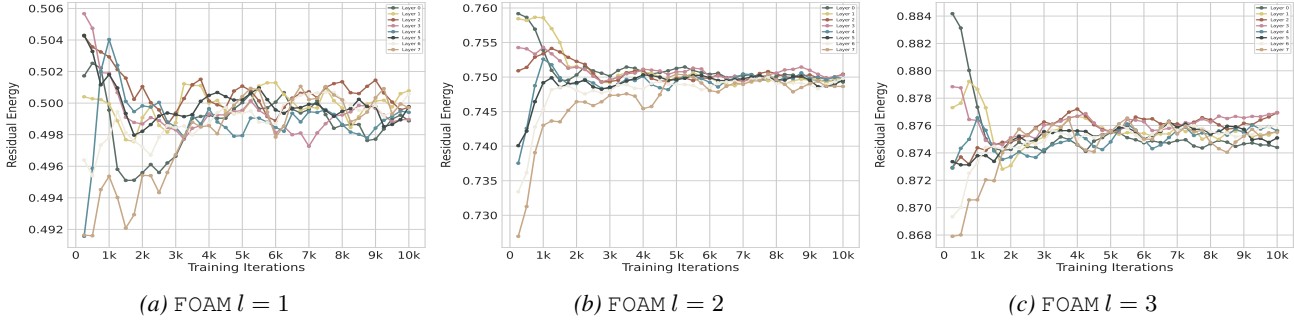

*(a)* FOAM $l = 1$        *(b)* FOAM $l = 2$        *(c)* FOAM $l = 3$

*Figure 10.* **The variation of residual energy ratio throughout training with different fold level $l$.** We report the average energy ratio across all modules within each layer. It can be observed that the energy ratio of the residual increases as $l$ grows, with the values concentrating around $1 - \frac{1}{2^l}$. This implies that the residual $R_t$ captures most of the energy from the original gradient, highlighting the necessity of injecting residuals.

*Table 14.* Recorded memory overhead for different methods on LLaMA-1.3B with a batch size of 32. We train the LLaMA-1.3B model on 32 NVIDIA RTX 3090 (24GB) GPUs.

| Methods | Memory | Methods | Memory |
|---------|--------|---------|--------|
| Full-Adam | 20.61G | GaLore-1/4 | 17.55G |
| Muon | 18.30G | APOLLO-1/4 | 17.56G |
| Adam-mini | 18.05G | FOAM-2 | 17.25G |
| GaLore-1/8 | 16.74G | GWT-Mini | 16.00G |
| APOLLO-1/8 | 16.74G | APOLLO-Mini | 16.00G |
| FOAM-3 | 16.57G | FOAM-Mini | 16.00G |

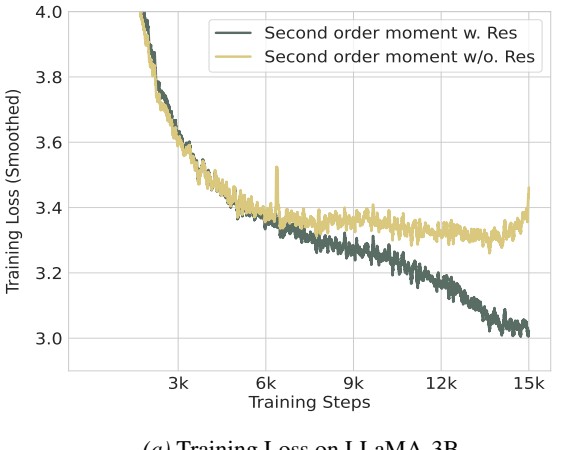

*(a)* Training Loss on LLaMA-3B

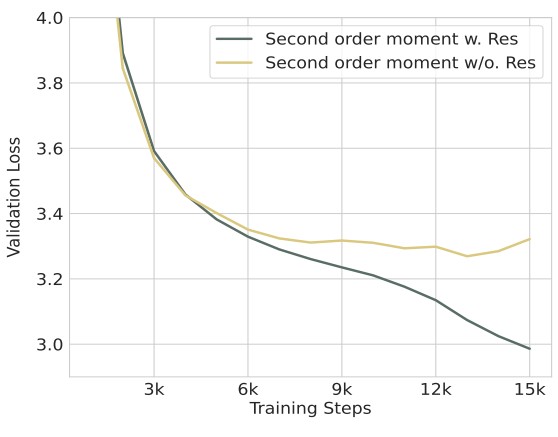

*(b)* Validation Loss on LLaMA-3B

*Figure 11.* **Ablation study of $R_t^2$ on the second-order moment.** The results show that incorporating residuals into the second-order moment leads to a more stable decrease in the training curve. In contrast, without residuals, the training exhibits a faster initial decrease but experiences a rise in loss during later stages.

## E. Benchmark Details

In this work, we evaluate our methods using several widely adopted benchmark datasets that cover both pre-training and downstream tasks.

- **C4 (Colossal Clean Crawled Corpus):**

  C4 is a large-scale English-language text corpus derived from Common Crawl data. It has been widely used for language model pretraining due to its scale and linguistic diversity. We follow the preprocessing and filtering steps introduced by the T5 framework to remove boilerplate and low-quality content.

- **MMLU (Massive Multitask Language Understanding):**

  MMLU is a comprehensive benchmark covering 57 tasks across various domains, designed to evaluate the reasoning and world knowledge of language models in a zero-shot setting. These tasks span multiple categories, including: STEM (e.g., physics, chemistry, mathematics), Humanities (e.g., history, philosophy, art), Social sciences (e.g., economics, psychology, political science), Other professional and academic subjects (e.g., law, computer science, clinical knowledge)

- **GLUE (General Language Understanding Evaluation):**

  GLUE consists of nine NLU tasks that test a model's general language understanding capabilities: CoLA (linguistic acceptability) (Warstadt et al., 2018), STS-B (semantic textual similarity) (Cer et al., 2017), MRPC (paraphrase detection) (Dolan & Brockett, 2005), RTE (recognizing textual entailment), SST-2 (sentiment analysis) (Socher

et al., 2013), MNLI (multi-genre natural language inference) (Williams et al., 2017), QNLI (question-answering NLI) (Rajpurkar et al., 2018), QQP (duplicate question detection). The broad coverage makes GLUE a standard benchmark for evaluating pre-training BERT models (Hu et al., 2022).

