# OpenReview forum: "FOAM: Blocked State Folding for Memory-Efficient LLM Training"
_ICML.cc/2026/Conference — ICML 2026 regular_

### Official Review · Reviewer_L8kd · 2026-03-12

**Soundness:** 2
**Presentation:** 3
**Significance:** 2
**Originality:** 3
**Overall Recommendation:** 4
**Confidence:** 4

**Summary:**

When training large language models with adaptive optimizers, there is a significant static memory overhead added through the necessity to store two optimizer states. Recent methods such as GaLore have attempted to address this concern by performing low-rank optimization, down-projecting the gradient signal using an SVD-based projection matrix. However, this adds computational overhead due to computing the SVD every $k$ steps, as well as additional memory requirements to store the projector (although this is amortised with the lower memory overhead). Instead, this paper introduces FOAM, an optimizer that compresses optimizer states by computing block-wise gradient means alongside a residual correction mechanism (error feedback). By avoiding low-rank projections, this approach eliminates the computational overhead relative to GaLore , while offering near-parity with full-rank optimization and surpassing existing memory-efficient baselines.

**Compliance With Llm Reviewing Policy:**

Affirmed.

**Final Justification:**

In my initial review, I assigned a 'Weak Reject' rating. While I found the paper to be well-executed, I was concerned about the absence of key ablation experiments, which left ambiguity as to why the performance gaps between FOAM and other memory-efficient baselines were so large. However, the authors provided convincing new results in their rebuttal, demonstrating that FOAM still outperforms the additional memory-efficient baselines that I had suggested. Given that the authors have addressed my primary concern regarding the completeness of their empirical evaluation, this has improved my assessment of the work and I have upgraded my final rating to a 'Weak Accept'.

**Key Questions For Authors:**

1. Why would SVD projection not be a superior method in terms of preserving the specific information necessary for optimization compared to your folding method? SVD explicitly preserves the principal directions of maximum variance, whereas this is not mathematically guaranteed by the folding operator. A more detailed theoretical discussion on this point would be appreciated.
2. The results indicate that Muon's improvements are relatively modest, whereas state-compression methods like FOAM—and to a lesser extent, low-rank optimizers like GaLore—yield more significant gains. Does this suggest that compressing optimizer momentum is a strictly superior approach to both full-rank adaptive optimizers like Adam and orthogonalized gradient methods like Muon for LLM training?
3. FOAM utilizes an instantaneous residual correction ($R_t$) that is applied to the unfolded update and immediately discarded. How does this memoryless approach compare theoretically and empirically to traditional persistent error feedback mechanisms (e.g., [1, 2]) that explicitly carry compression errors forward in time? Was a persistent error buffer ablated during your research, and if so, did the memory cost outweigh the performance gain?
4. Given that LDAdam [1] also addresses optimizer state compression using low-rank gradients, accounting for momentum rotation, combined with persistent error feedback, a comparison between FOAM and LDAdam would significantly strengthen the paper. In its current form, GaLore might be considered a weaker baseline because it lacks the mechanisms to account for compression loss and the rotating subspace issue that LDAdam solves. Could the authors provide this comparison or discuss the theoretical tradeoffs?
5. Observing the ablation study in Figure 11, the removal of the residual connection causes a severe degradation (stagnation) in performance. Mathematically, it appears that without the residual correction $R_t$, the update vectors are strictly constrained to the column space of the expansion operator $E^{(l)}$. Consequently, the parameter trajectory is locked into an affine subspace of dimension $d/l$, meaning the relative differences between weights within the same block can never change from their initialized values. Is this theoretical subspace constraint the primary cause of the stagnation observed in Figure 11? Elaborating on this mathematical limitation in the text would strongly motivate the necessity of the residual term when applied to the momenta terms.

**Limitations:**

Yes.

**Strengths And Weaknesses:**

Generally, this is a well executed paper both experimentally and in terms of its presentation. However, I think the research misses out key references in the literature. As such, these references introduce key ablation experiments that could potentially explain why the gaps between it and other memory-efficient baselines against which it compares are so large. My overall stance would be to encourage the authors to include these discussions in their work as without this additional context, the experiments are not thoroughly investigated and could artificially inflate the relative performance of their method.

**Presentation**

- **Strengths:** The paper is very well written. The authors have taken effort to provide both pictorial and mathematical evidence to illustrate the main contributions of the paper. Of the baselines and relevant literature it considers, it is able to show clear advantages in support of its method.
- **Weaknesses:** The authors fail to discuss or compare against highly relevant prior works in state-compression and error feedback [1,2]. This detracts from the overall contextualization of the paper, as comparing i) FOAM's memoryless residual to persistent error feedback mechanisms and ii) LDAdam as an improvement over GaLore (discussed more below) is crucial for evaluating its relative efficacy.

**Soundness**

- **Strengths:** The experimental settings (referring to the models, datasets, etc.) are well-chosen. The ablations of FOAM (in the context of the work they have chosen to reference) are well explained by both empirical and theoretical results.
- **Weaknesses:** The paper lacks rigor in theoretically contrasting FOAM's block-averaging with SVD-based methods. It is not entirely clear why the folding operator would be superior to SVD in preserving the most important variance directions necessary for optimization. Furthermore, the exclusion of LDAdam [1] as a baseline weakens the evaluation. LDAdam utilizes low-dimensional gradient statistics while accounting for the continually rotating low-rank subspace encountered throughout training, combined with persistent error feedback. FOAM instead relies on block-averaging with an instantaneous, memoryless residual. Without an empirical or theoretical comparison to methods that explicitly track compression errors across time, it is difficult to ascertain whether FOAM's instantaneous approach is truly optimal or if the performance gains over GaLore could simply be achieved by adding LDAdam-like improvements (error feedback + momentum rotation) to existing baselines. Furthermore, there is a question of the computational overhead that is addressed by LDAdam by moving to the power iteration that should be addressed to motivate the lack of SVD in this case.

**Significance**

- **Strengths:** This paper has high potential significance. It presents a novel framework for memory-efficient optimization, demonstrating that near full-rank performance can be achieved without the computational overhead of SVD-based solutions.
- **Weaknesses:** The true significance hinges on how FOAM holds up against state-of-the-art error-feedback memory-efficient methods like LDAdam. Resolving this comparison would clarify whether block-folding or low-rank projections offer superior performance in this regime (in effect, this would be a potential Pareto tradeoff).

**Originality**

- **Strengths:** The formulation of the folding operator combined with the instantaneous residual connection is an interesting and novel approach to avoiding the SVD bottleneck. The mechanism by which full-parameter updates are preserved via the residual is clever.
- **Weaknesses:** The originality of the core mechanism is slightly dimmed by the lack of discussion differentiating it from generalized error feedback. While FOAM's residual is memoryless, the broader concept of utilizing uncompressed residuals to correct compressed optimizer states is well-established [1, 2]. A deeper theoretical discussion explicitly distinguishing FOAM's instantaneous correction from persistent error tracking would strengthen the paper's novelty claims. Furthermore, a larger discussion on comparing how information is preserved (principal directions) by SVD versus block folding is crucial to improving the paper's positioning against prior work.

**References**

[1] Robert et al., LDAdam: Adaptive Optimization from Low-Dimensional Gradient Statistics

[2] Seide et al., 1-bit stochastic gradient descent and its application to data-parallel distributed training of speech DNNs

---

> ### Author Rebuttal · Authors · 2026-03-30
>
> ## Response to Reviewer L8kd
>
> We thank the reviewer for the insightful suggestion. In the following, we will discuss Error Feedback (EF) methods such as LDAdamW and the compression method such as SVD to address the weaknesses.
>
> ### Comparison with LDAdamW
>
> **Efficiency Comparison**
>
> We compare the memory and computational complexity of these methods with gradient $G\in \mathbb{R}^{m\times n}$ with $m \leq n$. FOAM performs compression via element-wise block averaging, whereas GaLore relies on SVD, and LDAdamW employs power iteration, rotation, and subspace projection.
>
> | |Memory|Complexity|
> |-|-|-|
> |FOAM|$mn/2^{l-1}$|$\mathcal{O}(mn)$|
> |LDAdamW|$mr+2rn$|$\mathcal{O}(mnr)$|
> |GaLore|$mr+2rn$|$\mathcal{O}(mn^2)$|
>
> Since FOAM does not involve matrix multiplications, it achieves higher computational efficiency. Moreover, it does not require storing projection matrices.
>
> **EF**
>
> FOAM adopts an instantaneous error correction, whereas LDAdamW employs tracked EF. The advantage of instantaneous error correction is that it incurs no additional memory overhead. LDAdamW reuses p.grad to store the error, but this design prevents the use of Gradient Clipping (GC) (GC is crucial for all tested optimizers when pre-training 7B models in our experiments).
>
> **Theoretical Justification**
>
> Unlike SVD, our block-averaging approach preserves locality and can be interpreted as compressing gradients from a frequency-domain perspective. From an information-theoretic standpoint, low-frequency components are typically more stable and less susceptible to noise than high-frequency components. In contrast, although SVD preserves directions of large variance, these directions may still be dominated by noise [He et al. Subspace, ICML 2025].
>
> **Experimental Results**
>
> For a comprehensive comparison, we adopt LDAdamW as a new baseline, pre-train models ranging from 60M to 1.3B, and evaluate lr in $5e-4, 1e-3, 2.5e-3, 5e-3$ with EF; hyperparameters follow the authors’ recommendations ($\beta_1=0.908, \beta_2=0.99, \rho=0.908$), set $r$ as $1/4$ of hidden size, and the remaining settings are consistent with Table 2 in the main text (owing to constraints in time and computational resources, we will add more comparsions of LDAdamW and set as a main baseline in the final version). We compare memory, PPL, and training time.
>
> | |60M|130M|350M|1.3B|
> |-|-|-|-|-|
> |GaLore|34.38(0.28G,0.95h)|26.47(0.57G.1.97h)| 19.36(1.38G,10.38h)|15.66(4.76G,46.21h)|
> |LDAdamW|29.27(0.27G,0.91h)|22.86(0.57G,2.28h)|17.53(1.38G,12.76h)|14.88(4.76G,49.73h)|
> |FOAM|28.53(0.27G,0.85h)|22.51(0.54G,1.90h)| 15.87(1.30G,9.63h)|13.13(4.45G,38.61h)|
>
> Our results reveal that FOAM achieves lower PPL and faster training speed while maintaining lower memory usage. Together with the experiments in the main text, this justifies our folding method as a promising alternative to SVD.
>
> ### Response to Questions (Q)
>
> **Q1: Discussion on block-folding**
>
> See above in **Theoretical Justification**
>
> **Q2: Compression as a superior method**
>
> The results do not necessarily imply strict superiority but reflect the intrinsic redundancy of LLM gradients. While Muon attempts to orthogonalize the full rank gradient, it may inadvertently whiten low variance noise. This phenomenon where compression or low rank projection yields superior optimization and generalization has been observed in other compression methods as well [Zhang et al. APOLLO, MLsys 2025; Chen et al. Fira, NeurIPS 2025]. Our findings suggest that for LLMs, filtering redundant dimensions via compression can be a more robust strategy for generalization than decorrelating all dimensions through orthogonalization in some contexts.
>
> **Q3: Instantaneous EF**
>
> Theoretically, traditional Persistent EF [Seide et al., 1-bit] aims to ensure unbiased updates in distributed settings, whereas LDAdamW compensates for information loss caused by low-rank projections by tracking accumulated errors in a rotated subspace. However, FOAM’s residual $R_t$ represents spatial high-frequency components filtered out by block-averaging. During early research, we ablated a persistent buffer and found it yielded marginal PPL gains (sometimes worse) and a larger memory footprint.
> | |60M|130M|
> |-|-|-|
> |FOAM+instantaneous EF|28.53(0.27G)|22.51(0.54G)|
> |FOAM+persistent EF|28.49(0.38G)|22.70(0.79G)|
>
> Our results reveal that, at least under FOAM’s block-folding strategy, instantaneous errors perform better than accumulated errors, which justifies that in LLM training, high-frequency components vary more drastically. Based on this, we adopted this design. We will include this in our revised version.
>
> **Q4: Comparison with LDAdamW**
>
> See above in **Comparison with LDAdamW**.
>
> **Q5: Necessity of Residual**
>
> The reviewer’s analysis is correct. Without the residual correction, the update is strictly restricted to a subspace of dimension $d/2^l$. We will incorporate this analysis into the revised manuscript to further motivate the necessity of our residual mechanism.

---

> > ### Author Rebuttal · Reviewer_L8kd · 2026-04-01
> >
> > Dear Authors,
> >
> > Thank you for the the time that you have taken to respond to my initial review. In light of your responses, I have adjusted my score accordingly. As a suggestion for the theoretical justification on SVD vs folding that you have provided: while this is discussed in prior work as you have cited, it would be good to see this shown mathematically and/or as a larger discussion in the main text and not just a citation to relevant work. I think this would improve the framing of the work relative to the baselines I had suggested.

---

> > > ### Author Response · Authors · 2026-04-03
> > >
> > > We are sincerely grateful for your constructive feedback and recognition of our efforts. We appreciate your suggestion regarding the theoretical discussion of the relevant work and will ensure that we include a detailed discussion in the final version.

---

### Official Review · Reviewer_JPL3 · 2026-03-13

**Soundness:** 3
**Presentation:** 2
**Significance:** 3
**Originality:** 2
**Overall Recommendation:** 4
**Confidence:** 3

**Summary:**

This paper introduces an optimization method that applies a dynamic state folding mechanism to the momentum, alongside a residual term, to reduce the memory footprint of optimizer states. Conceptually, the approach shares similarities with Adam-mini, but it replaces static row-wise operations with dynamic folding and also introduces a residual term. The authors provide experiment results and a theoretical convergence guarantee to support the performance of the algorithm.

**Compliance With Llm Reviewing Policy:**

Affirmed.

**Final Justification:**

I think the authors' rebuttal basically addressed my concerns in the theoretical part. And the rebuttal also presents some ablation study and understanding. The major weakness for me is still the novelty, since folding is a known technique. But the results look firm and I will say this is a solid paper in general.

**Key Questions For Authors:**

1. Is there a specific necessity to define the dimension as $2^l$? It conceptually appears that any arbitrary dimension size could be viable. Does this restriction stem from specific hardware efficiency considerations, or is it purely a design choice?
2. Is it possible that the empirical success is primarily driven by the residual term rather than the folding mechanism itself? Based on the results in around row 403, Adam-mini also performs well when FOAM is integrated. How about we directly employ the residual term and test whether it also improves Adam-mini?
3. FOAM performs even better than AdamW and Muon is surprising, given that it is a memory focus algorithm. Could you provide more intuition on why this occurs? Furthermore, it would be interesting to explain why this method shows great improvements when applied to Adam-type optimizers but seemingly degrades performance when applied to Muon.

**Strengths And Weaknesses:**

**Strengths**:
1. The motivation and considered approach are good and meaningful.
2.  The experimental evaluation is comprehensive, and the settings seem to be generally fine. The consistent performance gains observed across different setups indicate the method's practical benefits.
3. I think the novelty and benefits of the method mainly come from the residual term. Although it is similar to the error feedback, it is the first time I have seen it in memory-efficient algorithms.

**Weakness**:
1. The block-folding thing mentioned in the title and algorithm name is actually very similar to Adam-mini. The difference is that Adam-mini considers the folding based on structural observations, but FOAM considers the folding by tuning it manually. I think the authors may consider changing the focus of paper writing.
2. The convergence theorem does not present good enough results. For an algorithm with a decaying step size, the bound exhibits a direct, non-decaying dependence on the variance $\sigma^2$. A standard optimization analysis should yield a decaying variance term (e.g., scaling with $\sigma/\sqrt{T}$), meaning the current theorem fails to match conventional convergence rates. Also, we don't need to include the term $\delta_l$ in the bound if $\delta_l < 1$.
3. The mathematical formulation in Section 3.2 is awkwardly presented. For instance, the dimensional representation of matrix $A$ (which should be $n \times n$ based on the mathematical formulation) is written in a way that is neither mathematically rigorous nor intuitive, making it difficult for readers to follow.

---

> ### Author Rebuttal · Authors · 2026-03-30
>
> ## Response to Reviewer JPL3
> We sincerely thank the reviewer for the constructive feedback and for recognizing the practical benefits and novelty of our work. We address the identified Weaknesses (W) and Questions (Q) as follows:
>
> **W1: Relationship with Adam-mini**
>
> The term folding reflects the compression’s conceptual resemblance to folding operations. While FOAM shares some similarities with Adam-mini, it differs in three key aspects: (1) Granularity: FOAM offers finer-grained folding control while Adam-mini compresses row-wise or module-wise; (2) Scope: FOAM compresses both first and second moments, whereas Adam-mini focuses only on the second; (3) Residuals: FOAM incorporates a critical residual term absent in Adam-mini. We will clarify these structural distinctions in the revised version.
>
> **W2: Improved Convergence Theorem**
>
> The variance term in the theorem arises from an overly conservative scaling of $||\Delta_t||^2 = ||M_t - \nabla f(W_t)||$. In the original Lemma A.2, the definition of $S_t$ (Eq. 8) introduced a constant $\sigma^2$ during the estimation of $||\Delta_t||^2$. To address this, we have revised the proof by decomposing $\Delta_t$ as $\Delta_t = \hat{\Delta}\_t + \Xi_t$, isolating the noise-related component $\Xi_t$. By deferring the treatment of $\Xi_t$ to the main theorem, we eliminate the constant variance term. Therefore, we can rewrite the inner product term in Lemma A.3 as
> $$\left\langle\nabla f(W_t),\frac{\Delta\_t}{\sqrt{V_t}+\epsilon}\right\rangle=\left\langle\nabla f(W_t),\frac{\hat{\Delta}\_t+\Xi_t}{\sqrt{V_t}+\epsilon}\right\rangle$$ This allows us to handle the variance term in the main theorem, ultimately leading to the following conclusion.
> $$\min_{1\le t\le T}\mathbb{E}[||\nabla f(W_t)||^2] = \mathcal{O}\left(\frac{(1+\sigma^2)\log T}{\sqrt{T}}\right)$$
>
> **W3: Mathematical Formulation (Sec 3.2)**
>
> We apologize for any confusion caused by the notation. In the revision, Section 3.2 will be rewritten using rigorous tensor notation to ensure all parameter dimensions are unambiguous. To improve intuition, we will also include pseudo-code representations (e.g., explicit broadcasting functions) to clearly illustrate the compression operations.
>
> **Q1: Necessity of $2^l$ Dimension**
>
> While FOAM supports arbitrary folding factors, choosing $2^l$ is a strategic design for Transformer architectures. LLM hidden dimensions are typically powers of 2 (or multiples thereof), and using $2^l$ minimizes unnecessary padding. Moreover, this choice allows efficient utilization of hardware-accelerated kernels, such as PyTorch’s `repeat_interleave`, which is optimized for power-of-2 expansions.
>
> **Q2: Impact of Residuals on Adam-mini**
>
> In our paper, Adam-mini+FOAM denotes combining Adam-mini’s second-moment compression with FOAM’s first-moment folding (including the residual term). Since FOAM’s residuals originate from first-moment compression ($res:=G-GAE$), and Adam-mini lacks a first-moment compression mechanism, isolating the residual’s impact requires defining a custom residual for Adam-mini. Specifically, Adam-mini’s second-moment compression uses $\mathbb{E}[G^2]$. To generate a corresponding residual, we define $res:=(G-\mathbb{E}[G])^2$, which preserves positivity and avoids negative values under the square root when incorporated into the second moment (note that $G^2 - (\mathbb{E}[G])^2$ is not applicable for this reason). This residual is then injected into the second-moment update. To ensure fair evaluation, we test learning rates in $5e-4, 1e-3, 2.5e-3, 5e-3$. We present the results in the table below.
> | |LLaMA-60M|LLaMA-130M|
> |-|-|-|
> |FOAM|28.53|22.51|
> |Adam-mini|29.63|23.73|
> |Adam-mini+res|30.78|23.84|
>
> Interestingly, our results indicate that residual compensation can lead to performance degradation, in contrast to the significant gains observed with FOAM. We attribute this difference to the granularity of the compression mechanism. FOAM employs fine-grained block-folding, which preserves local spatial consensus. In this setting, the residual captures high-frequency components that align well with the update direction. By contrast, Adam-mini’s row/module-wise compression results in a residual that is overly noisy and less informative. This confirms that our empirical success is jointly attributed to the folding mechanism and the residual term.
>
> **Q3: Performance Gains over AdamW and Degradation on Muon**
>
> Intuitively, the performance improvements of FOAM over AdamW likely arise from its smoothing effect, which filters gradient noise while incorporating spatial information from neighboring parameters, thereby promoting structured updates rather than purely element-wise adjustments. Conversely, the observed performance degradation on Muon suggests that mean-based folding, which shares momentum across adjacent elements, conflicts with Muon’s coordinate-sensitive design and disrupts its orthogonalization mechanism. We will include this analysis in the revised discussion section.

---

> > ### Author Rebuttal · Reviewer_JPL3 · 2026-04-03
> >
> > Thanks for the detailed reply and the hard work. I think my questions are basically resolved, especially in the theory part. Although the novelty is still somewhat limited since the folding technique is not a very new one, but the results look promising to me and the ablations are good. I will increase my score to 4.

---

### Official Review · Reviewer_G3JF · 2026-03-15

**Soundness:** 3
**Presentation:** 3
**Significance:** 4
**Originality:** 4
**Overall Recommendation:** 4
**Confidence:** 4

**Summary:**

The paper introduces FOAM (Folded Optimizer with Approximate Moment), a novel framework designed to alleviate the memory bottlenecks associated with training Large Language Models (LLMs) using optimizers like Adam. By employing a blocked averaging technique (state folding) combined with a residual correction mechanism, the method reduces optimizer memory overhead by up to 90% and total memory footprint by up to 50% without sacrificing convergence speed or accuracy. Overall, the article addresses an important concept in high-performance deep learning. The authors claim to present an important context for scaling model training on hardware with limited VRAM by demonstrating superior performance across various benchmarks compared to existing low-rank or projection-based methods.

**Compliance With Llm Reviewing Policy:**

Affirmed.

**Key Questions For Authors:**

1. The study is constrained by computational resources to a maximum model size of 7B parameters, meaning its effectiveness and stability at the "ultra-large" scale of hundreds of billions of parameters remains to be verified on massive GPU clusters.
2. The current implementation of the folding operator is specifically designed for 2D weight matrices, requiring further architectural design to effectively handle the high-dimensional gradients often found in non-standard or experimental layer types.
3. While the paper mentions compatibility with gradient segmentation methods like ZeRO, it does not provide a detailed quantitative analysis of the inter-GPU communication overhead or the specific scaling laws when integrated with distributed system-level optimizations.

**Limitations:**

see above

**Strengths And Weaknesses:**

Strengths
1. The method achieves up to 90% reduction in optimizer memory overhead while maintaining the ability to perform full-parameter updates, unlike parameter-efficient methods like LoRA that restrict updates to a low-rank subspace.
2. FOAM incorporates a unique residual correction term that recovers information lost during the folding process, ensuring that individual parameter updates remain differentiated and avoiding the instability of simple block-sharing schemes.
3. The authors provide a rigorous theoretical proof demonstrating that FOAM maintains the same convergence rate as vanilla Adam  under standard non-convex optimization settings.
4. Extensive empirical validation across different model families (LLaMA, Qwen, RoBERTa) and tasks (pre-training and fine-tuning) shows that FOAM consistently outperforms strong baselines like GaLore, APOLLO, and Adam-Mini in terms of validation perplexity.
5. The optimizer-agnostic nature of the framework allows it to be seamlessly integrated with other advanced optimizers such as Muon and Adam-Mini, showcasing its versatility as a "plug-and-play" solution.
6. Unlike SVD-based methods such as GaLore, FOAM eliminates the need for expensive projection matrix computations, leading to higher training throughput and reduced computational overhead.

Weaknesses
1. Although the method introduces a fold level hyperparameter, there is limited exploration into how the optimal choice of might vary across significantly different architectural designs or extremely heterogeneous hardware setups.
2. The empirical results show a slight degradation in perplexity as sequence length increases, suggesting that the compression might be more sensitive to context length than standard Adam in long-range dependency tasks.
3. The evaluation is primarily focused on standard Transformer-based language models, leaving the effectiveness of the blocked averaging technique on other architectures like Diffusion Models or Vision Transformers largely unexplored.
4. While the residual term is shown to be critical for convergence, the added step of computing the residual at every iteration introduces a small amount of additional arithmetic operations compared to a purely shared-state approach.

---

> ### Author Rebuttal · Authors · 2026-03-30
>
> ## Response to Reviewer G3JF
>
> We thank the reviewers for their constructive feedback and high evaluation of FOAM's originality and significance. We address the identified Weaknesses (W) and Questions (Q) as follows:
>
> **W1: Robustness of Fold Level $l$**
>
> This hyperparameter is analogous to $r$ in low-rank methods, with $l$ mainly governing the compression intensity. A larger $l$ lowers memory usage (with an optimizer memory of $mn/2^l$) but can cause slight performance drops. We supplement this with experiments training LLaMA, Qwen, and GPT on the C4 dataset while varying $l$.
>
> | |l=2|l=3|l=4|l=5|l=6|l=7|
> |-|-|-|-|-|-|-|
> |LLaMA-130M|22.51|22.58|22.66|22.80|22.84|22.91|
> |Qwen-130M|22.47|22.56|22.62|22.71|22.83|22.88|
> |GPT-125M|24.66|24.79|24.91|25.03|25.07|25.23|
>
> Our experimental results confirm that across different model architectures, the effect of $l$ is consistent, with smaller $l$ achieving better performance. This allows $l$ to focus on balancing performance and memory without the need for model-specific tuning.
>
> **W2: Performance on Long Sequences**
>
> As shown in Table 7 in the main text, performance degradation under long-sequence settings is common, with the PPL of all tested optimizers increasing compared to shorter sequences. This is related to our experimental setup: to ensure comparability, we keep the total number of tokens per batch constant. A larger sequence length reduces the batch size, so performance degradation in this context is entirely expected. Nevertheless, under this experimental setting, FOAM still achieves lower PPL compared to the other tested baselines.
>
> **W3: Generalization Beyond Transformers**
>
> We chose transformer models mainly to facilitate comparison with baselines; all memory-efficient baselines in this work are designed for transformers and have been widely applied to transformer architectures. While FOAM is inherently generalizable, extending it to other models, like diffusion models, necessitates designing a new FOAM compression strategy and adapting the baselines accordingly. This point has already been clarified in the "Future Works" section of the main text.
>
> **W4: Computational Overhead of Residuals**
>
> Residual computation is purely element-wise, incurring negligible overhead compared to forward/backward passes. To confirm this, we report the PPL, iteration counts, and token throughput from pre-training a 3B model. We observe that FOAM with residual correction maintains nearly the same token throughput as FOAM without residual correction, differing by less than 1%. FOAM without residual correction, however, cannot converge, and the loss diverges entirely. This validates the key role of our residual correction, and even with it, our approach maintains faster training speed than Adam.
>
> | |PPL|Second/iteration|Token/s|
> |-|-|-|-|
> |Adam-8bit|14.19|3.79|34.52K|
> |FOAM with Res|11.98|3.66|35.77K|
> |FOAM without Res|Loss diverge|3.65|35.91K|
>
> **Q1: Scaling to ultra-large (100B) Models**
>
> Due to constrained computational resources, training ultra-large models (e.g., 100B parameters) remains beyond our current reach. Such large-scale training is generally prohibitive in academic settings and is typically reserved for industrial-scale infrastructures. The effectiveness of FOAM arises from the local redundancy and low-rank properties of gradients, which become more pronounced in larger models. Moreover, because FOAM’s compression–decompression operations are element-wise, its computational overhead grows linearly with model size. These properties allow FOAM to scale to training large models, a point we have also clarified in the "Future Works" section. We believe that the extensive experiments in our work validate FOAM as an effective compression strategy beyond SVD and represent a promising direction for future exploration.
>
> **Q2: Support for High-Dimensional Gradients**
>
> Designing compression methods for 2D weights (linear layers like attention and MLP) is also a widely adopted strategy in the field (e.g., GaLore, APOLLO, Muon), as 2D weights constitute the majority of parameters in modern LLMs, and compressing them yields the most significant results. For high-dimensional tensors (e.g., Conv), FOAM can be extended via N-dimensional block folding, treating each dimension as a spatial axis for local averaging. We agree that a generalized N--dimensional folding framework would improve architectural robustness, and we plan to open-source a more flexible implementation for non-standard layer types.
>
> **Q3: Compatibility with ZeRO**
>
> Theoretically, FOAM is fully compatible with ZeRO because it only compresses optimizer states. However, full integration requires fine-grained alignment between FOAM’s blocking strategy and ZeRO’s partitioning logic, an engineering-heavy task. Since communication overhead is not the primary focus of this work, we prioritize establishing FOAM’s core efficacy and leave distributed system-level optimization for future work.

---

### Decision · Program_Chairs · 2026-04-30

**Decision:**

Accept (regular)

**Comment:**

The paper proposes FOAM, a memory‑efficient optimizer for LLMs that folds optimizer states via block‑wise averaging and recovers lost information through a residual correction term to accomplish reduction in optimizer‑state memory and total memory footprint while maintaining full‑parameter updates and convergence comparable to Adam. The method is optimizer‑agnostic, avoids expensive SVD‑based projections, and shows strong empirical gains over baselines such as GaLore, APOLLO, and Adam‑Mini.

The reviewers appreciated the paper for its well‑motivated and practical approach to optimizer‑state compression, strong empirical results across multiple LLMs, and the novel use of block‑wise state folding with residual correction, while encouraging a more thorough discussion and comparison with recent state‑compression and error‑feedback methods to further strengthen its positioning. Further, the reviewers pointed out that the theoretical justification for SVD versus folding can be strengthened by presenting it more explicitly in the main text, either through mathematical derivation or a more detailed discussion, rather than relying primarily on citations to prior work, for completeness and clarity.

As a result, the authors are also advised to include comparisons to the following works, including SubTrack++, which is known to outperform LDAdam and GaLore:

- Robert et al., “LDAdam: Adaptive Optimization from Low‑Dimensional Gradient Statistics”

- Seide et al., “1‑bit Stochastic Gradient Descent and Its Application to Data‑Parallel Distributed Training of Speech DNNs”

- Rajabi et al., “SubTrack++: Gradient Subspace Tracking for Scalable LLM Training,” NeurIPS, 2025